# Determining the probability of hemiplasy in the presence of incomplete lineage sorting and introgression

Mark S Hibbins[1]*, Matthew JS Gibson[1], Matthew W Hahn[1,2]

[1]Department of Biology, Indiana University, Bloomington, United States;
[2]Department of Computer Science, Indiana University, Bloomington, United States

**Abstract** The incongruence of character states with phylogenetic relationships is often interpreted as evidence of convergent evolution. However, trait evolution along discordant gene trees can also generate these incongruences – a phenomenon known as hemiplasy. Classic comparative methods do not account for discordance, resulting in incorrect inferences about the number, timing, and direction of trait transitions. Biological sources of discordance include incomplete lineage sorting (ILS) and introgression, but only ILS has received theoretical consideration in the context of hemiplasy. Here, we present a model that shows introgression makes hemiplasy more likely, such that methods that account for ILS alone will be conservative. We also present a method and software (*HeIST*) for making statistical inferences about the probability of hemiplasy and homoplasy in large datasets that contain both ILS and introgression. We apply our methods to two empirical datasets, finding that hemiplasy is likely to contribute to the observed trait incongruences in both.

## Introduction

Convergent evolution of the same phenotype in distantly related species provides some of the most compelling evidence for natural selection. Comparative inferences of convergence require that the species history is known (*Felsenstein, 1985*). Comparative methods applied to such histories implicitly assume that the loci underlying convergent traits also follow the species tree. However, gene trees at individual loci can disagree with each other and with the species tree, a phenomenon known as gene tree discordance. While genomic data allow us to overcome many technical sources of discordance (*Delsuc et al., 2005*; *Dunn et al., 2008*; *Misof et al., 2014*), discordance also has biological causes (*Degnan and Rosenberg, 2009*), and remains a common feature of phylogenomic datasets (*Pollard et al., 2006*; *Fontaine et al., 2015*; *Pease et al., 2016*; *Novikova et al., 2016*; *Wu et al., 2018*; *Vanderpool et al., 2020*).

Gene tree discordance can have multiple sources, including biological causes such as incomplete lineage sorting (ILS), introgression, and horizontal gene transfer, and technical causes such as hidden paralogy or errors in gene tree inference (*Schrempf and Szöllősi, 2020*). Here, we focus primarily on the first two biological causes: ILS and introgression. Looking backwards in time, ILS is the failure of lineages to coalesce within a population before reaching the next most recent ancestral population. The probability of discordance due to ILS is a classic result of the multispecies coalescent, and depends on the population size and the length of time in which coalescence can occur (*Hudson, 1983*; *Pamilo and Nei, 1988*). More recently, the classic multispecies coalescent model has been extended to include introgression (a term we use to encompass hybridization and subsequent gene flow), in a framework called the 'multispecies network coalescent' (*Yu et al., 2012*; *Yu et al., 2014*; *Wen et al., 2016*). In this model, species relationships are modeled as a network, with introgression represented by horizontal reticulation edges. Individual loci probabilistically follow or do

*For correspondence:
mhibbins@iu.edu

Competing interests: The authors declare that no competing interests exist.

not follow the reticulation edge, after which they sort according to the multispecies coalescent process (i.e. with ILS). A major advantage of this approach is that ILS and introgression can be modeled simultaneously (reviewed in *Degnan, 2018*), allowing for more detailed study of the consequences of discordance.

Importantly, discordant gene trees can lead to the appearance of apparently convergent traits. This is because discordant gene trees have internal branches that do not exist in the species tree. If a mutation occurs along such a branch at a locus controlling trait variation, it may produce a pattern of character states that is incongruent with the species tree. Incongruent trait patterns are the basis for inferences of convergent evolution ('homoplasy'), and thus this phenomenon has become known as hemiplasy (*Avise and Robinson, 2008*). Since hemiplasy can produce the same kinds of trait incongruence as homoplasy, failing to account for gene tree discordance can generate misleading inferences about convergence (*Mendes and Hahn, 2016*; *Mendes et al., 2016*). Studies in systems with widespread discordance have found that hemiplasy is a likely explanation for many patterns of incongruence (*Copetti et al., 2017*; *Wu et al., 2018*; *Guerrero and Hahn, 2018*).

The problem of hemiplasy makes it clear that robust inferences about the evolution of traits must account for gene tree discordance (*Hahn and Nakhleh, 2016*). Recent work has provided expressions for the probabilities of hemiplasy and homoplasy (*Guerrero and Hahn, 2018*), allowing for an assessment of whether a single transition (hemiplasy) or two transitions (homoplasy) is more likely to explain trait incongruence. This model shows that the most important factors contributing to a high risk of hemiplasy relative to homoplasy are a short internal branch on the species tree (which increases the rate of gene tree discordance), and a low mutation rate (which reduces the probability of the multiple independent transitions needed for homoplasy). However, applying this model in present form to empirical phylogenetic data faces two major limitations. First, incomplete lineage sorting is the only source of gene tree discordance considered, excluding scenarios with gene flow. Second, the model is limited to evolution along a three-taxon tree, restricting calculations for the exact probability of hemiplasy in larger clades.

With genomic data now available for many species, it has become clear that introgression is a common phenomenon (*Mallet et al., 2016*). Introgression leads to different patterns of gene tree discordance than expected under ILS alone – specifically, minority gene tree topologies supporting a history of introgression are expected to become more common than those produced via ILS alone. These differences form the conceptual basis for common tests of introgression using genomic data (*Reich et al., 2009*; *Green et al., 2010*; *Durand et al., 2011*; *Patterson et al., 2012*; *Pease and Hahn, 2015*). Introgression also affects the expected coalescence times between pairs of species (*Joly et al., 2009*; *Brandvain et al., 2014*; *Hibbins and Hahn, 2019*; *Hahn and Hibbins, 2019*). Pairs of species that have exchanged genes will have lower levels of sequence divergence, and therefore longer shared internal branches, at introgressed loci than expected under ILS alone. These differences in the frequency and branch lengths of genealogies produced by introgression should meaningfully affect the probability of hemiplasy. Therefore, it is important that both sources of gene tree discordance be accounted for in models of trait evolution.

For trees with more than three taxa, the number of possible gene trees and mutational configurations that could explain a particular pattern of trait incongruence increases dramatically. To illustrate this problem, we consider two cases of empirical incongruence of a binary trait. First, consider the case of New Guinea lizards that have evolved green blood from a red-blooded ancestor (*Figure 1A*; *Rodriguez et al., 2018*). A clade of 15 taxa contains both the green-blooded species and red-blooded species (the ancestral state). Given the phylogenetic distribution of the six green-blooded species—and no consideration of gene tree discordance—four independent transitions are necessary to explain this incongruence (*Figure 1*). However, the internal branches on this tree are short and discordance is likely. Individual loci could therefore group the green-blooded taxa into as few as one and as many as six separate clades. Depending on the history at loci affecting blood color, the distribution of green-blooded taxa could therefore be explained by anywhere from one to six mutations, and even more if we consider back-mutations. The one-mutation case represents a single transition due to hemiplasy along a branch that does not exist in the species tree, while the two- and three-mutation cases represent a combination of hemiplasy and homoplasy. The problem becomes even more complex when introgression occurs in the phylogeny, because each reticulation event introduces a new set of gene trees formed from the coalescent process at introgressed loci (*Hibbins and Hahn, 2019*). One such example is the origin of a chromosomal inversion spanning a

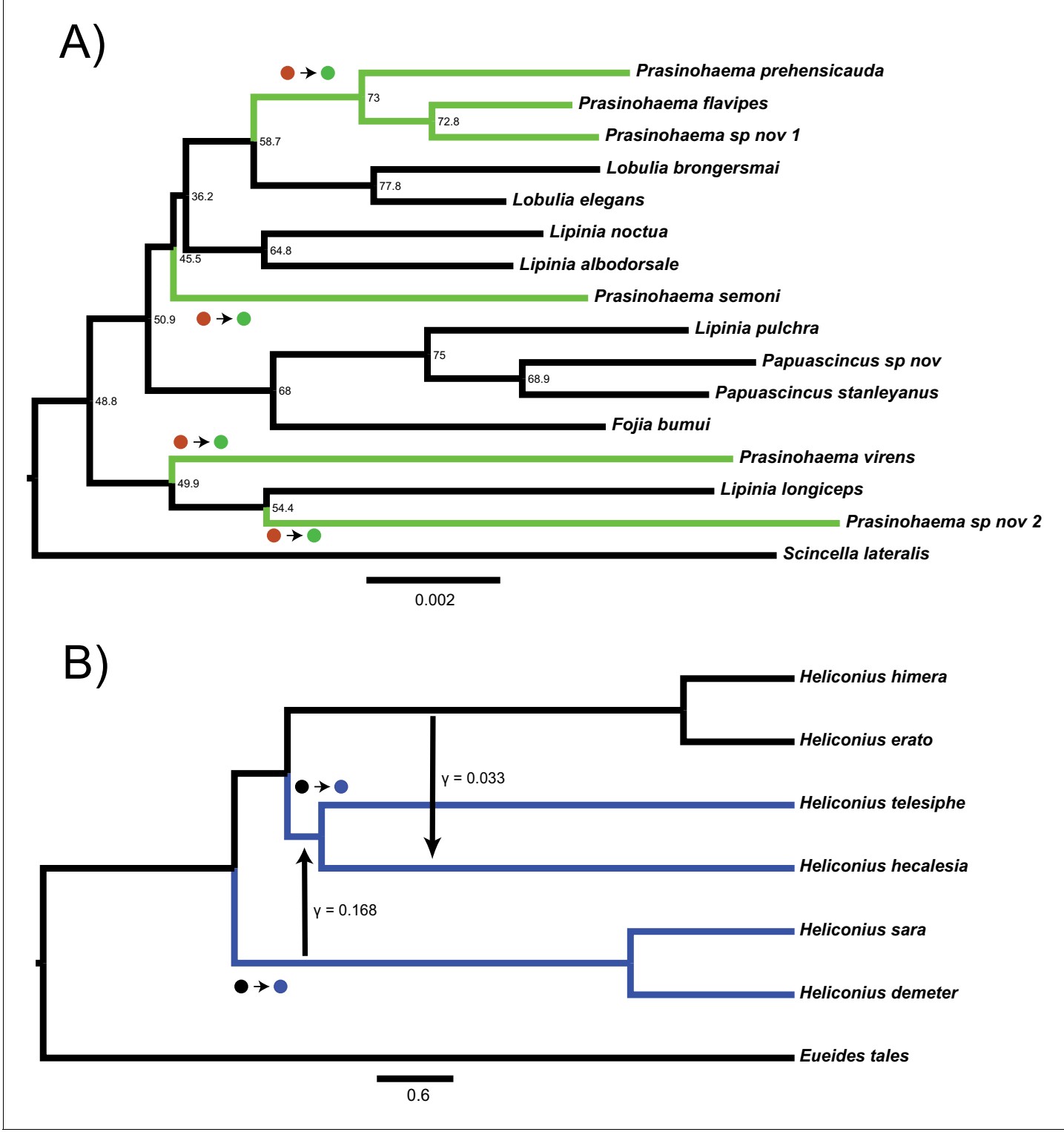

**Figure 1.** Two empirical examples of apparent convergence in character states that could potentially be explained by hemiplasy. (**A**) Maximum-likelihood species tree of the clade including green-blooded lizards and an outgroup, constructed from the concatenation of 3220 ultra-conserved elements (data from **Rodriguez et al., 2018**). Branch lengths in substitutions per site; nodes labeled with site concordance factors. (**B**) Coalescent network of *Heliconius erato/sara* clade, processed from the network constructed for the clade in **Edelman et al., 2019**. Branch lengths in units of 2*N* generations; rate, direction, and approximate timing of introgression events indicated by vertical arrows. In both trees, taxa with derived characters are colored, and the most parsimonious transitions from ancestral to derived states are labeled with circles.

*Figure 1 continued on next page*

*Figure 1 continued*

The online version of this article includes the following source data and figure supplement(s) for figure 1:

**Source data 1.** Input file given to *HeIST* for the lizard analysis, using the tree and character states shown in *Figure 1A*.
**Source data 2.** Input file given to *HeIST* for the butterfly analysis, using the tree, character states, and introgression events shown in *Figure 1B*.
**Figure supplement 1.** Full 43-species maximum-likelihood lizard phylogeny constructed by *RAxML*, with branch lengths in units of substitutions per site.
**Figure supplement 2.** Phylogeny of green-blooded lizards and an outgroup inferred from 3220 UCE gene trees using *ASTRAL* (data from *Rodriguez et al., 2018*).

gene involved in wing coloration in the *Heliconius erato/sara* clade of butterflies (*Figure 1B*; *Edelman et al., 2019*). Overall, the huge number of possible gene trees (>213 trillion for 15 lizard species; *Felsenstein, 2004*) and the large number of possible mutational events on these trees makes it infeasible to derive an explicit mathematical solution to address questions about hemiplasy in many empirical systems.

Here, we make two steps toward addressing these problems. First, we derive expressions for the probabilities of hemiplasy and homoplasy under the multispecies network coalescent for three taxa. Our results show that hemiplasy becomes increasingly likely relative to homoplasy as introgression occurs at a higher rate and at a more recent time relative to speciation. We also show how this pattern is influenced by the direction of introgression. These results highlight the need to account for both ILS and introgression in order to understand the origins of a trait incongruence. Second, we present a tool called *HeIST* (*H*emiplasy *I*nference *S*imulation *T*ool) that uses coalescent simulation to dissect patterns of hemiplasy and homoplasy in larger phylogenies. This tool provides an estimate of the most likely number of transitions giving rise to observed incongruence of binary traits, and accounts for both ILS and introgression. Lastly, we apply *HeIST* to two empirical cases of apparent convergence in a binary trait, finding that hemiplasy is likely to contribute to the observed trait incongruences.

## Results

### A model for the probability of hemiplasy under the multispecies network coalescent

To study the effects of introgression on the probability of hemiplasy, we combine concepts from two previously published models: the 'parent tree' framework of *Hibbins and Hahn, 2019*, and the model of binary-trait evolution presented in *Guerrero and Hahn, 2018* (see *Wang et al., 2020* for an alternative way to extend the model to incorporate introgression). Consider a rooted three-taxon tree with the topology ((A,B), C). We define $t_1$ as the time of speciation between lineages A and B in units of 2*N* generations, and $t_2$ as the time of speciation between C and the ancestor of A and B. We also imagine an instantaneous introgression event between species B and C at time $t_m$, which can be in either direction (C → B or B → C). We define the total probability of a locus following an introgressed history as $\delta$, with $\delta_2$ denoting the probability of C → B introgression, and $\delta_3$ the probability of B → C introgression. Introgression in both directions at an individual locus is not allowed in our model. However, a single introgression event in both directions can be modeled by allowing different directions at different loci. The history described here is represented by the phylogenetic network shown in *Figure 2* (top). Other introgression scenarios can be accommodated by our model (see Discussion), but will not be considered here.

To make it easier to track the history of different gene trees, we imagine that a phylogenetic network can be split into a set of 'parent trees' which describe the history at individual loci (*Meng and Kubatko, 2009*; *Liu et al., 2014*; *Hibbins and Hahn, 2019*; *Figure 2*, bottom). Within each of these parent trees, which describe either the species history or the history of introgression, gene trees sort under the multispecies coalescent process. Loci follow the species history, referred to as parent tree 1, with probability $1 - (\delta_2 + \delta_3)$. With C → B introgression, some loci will follow the alternative history within parent tree 2, with probability $\delta_2$. In parent tree 2, species B and C are sister and share a 'speciation' time of $t_m$. B → C introgression causes loci to follow parent tree 3 with probability $\delta_3$; in this history, lineages B and C are sister and split at time $t_m$, while the split time of A and the ancestor of

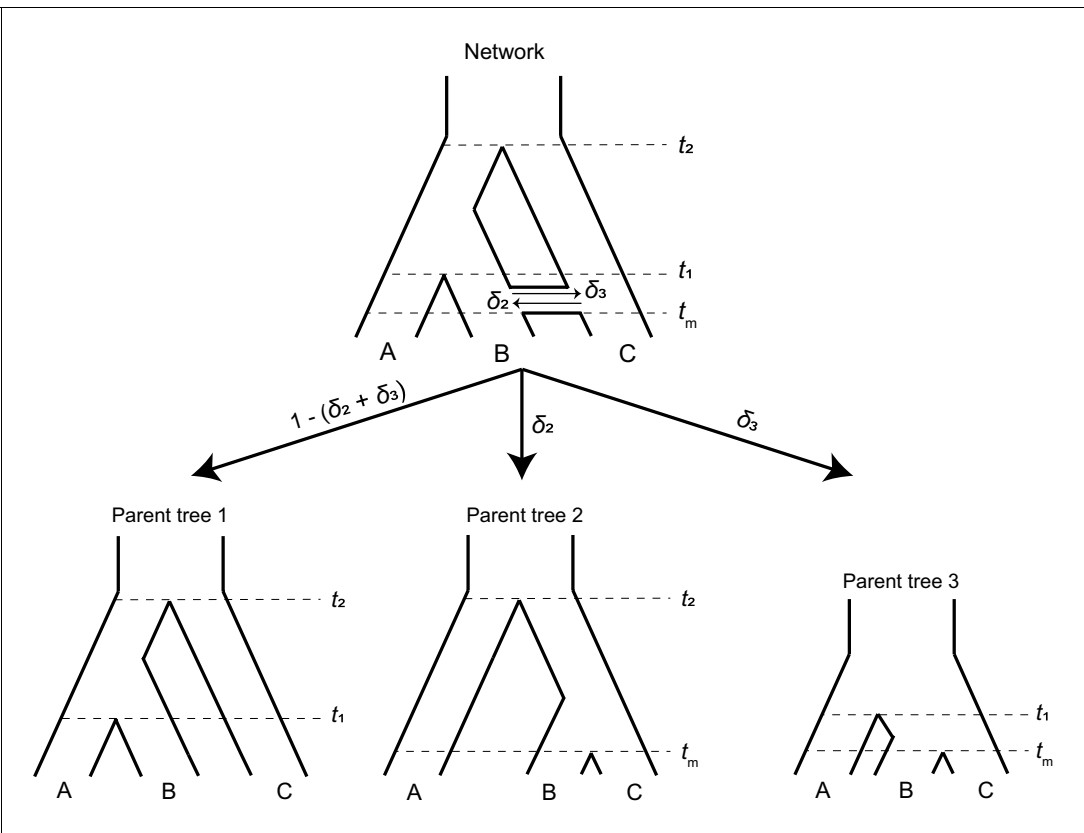

**Figure 2.** A phylogenetic network (top) can be split into a set of parent trees (bottom) representing the possible histories at individual loci. The probability that a locus is described by a particular parent tree depends on the probability of introgression (arrow labels). The horizontal 'tube' shown in the phylogenetic network does not depict introgression over a continuous time interval, but rather shows the timing of introgression ($t_m$) in an instantaneous pulse, while allowing for coalescence to be visualized for loci that follow a history of introgression.

The online version of this article includes the following figure supplement(s) for figure 2:

**Figure supplement 1.** Each parent tree in our model generates four gene trees: one generated from lineage sorting (Panels **A** and **E**), and three equally likely trees generated from incomplete lineage sorting (panels **B-D, F–H**).

B/C is reduced to $t_1$. This reduction in the second split time in parent tree 3 occurs because the presence of loci from lineage B in lineage C allows C to trace its ancestry through B going back in time. Since B is more closely related to A than C, this allows C to coalesce with A at an earlier time (*Figure 2*). Each introgression event is modeled as a discrete and instantaneous 'pulse' that generates its own parent tree, and in our model we consider a single introgression event for simplicity. However, multiple events or introgression over a continuous time interval can be modeled by introducing multiple pulses with different directions, timings, or probabilities. Each such event introduces its own parent tree and set of gene trees.

Each parent tree can produce four gene trees under the multispecies coalescent process: one tree from lineage sorting, and three equally probable trees from incomplete lineage sorting (*Figure 2—figure supplement 1*). In other words, introgression always involves ILS, as these are not mutually exclusive histories. Each of these possible gene trees has five branches along which mutations can occur: three tip branches, an internal branch, and an ancestral branch. A subset of these possible gene trees within each parent tree can lead to hemiplasy, while homoplasy can happen in any gene tree (*Figure 3*). *Guerrero and Hahn, 2018* provide exact expectations for the probability of a mutation on each branch of each genealogy in an ILS-only model. Before extending this framework to incorporate introgression, the ILS-only model will be briefly described here, using a slightly updated notation that will make it easier to include the effects of introgression.

Consider a binary trait that is incongruent with the described species tree, where species B and C have the derived state and A has the ancestral state. We denote $\lambda_1$, $\lambda_2$, and $\lambda_3$ as the tip branches in

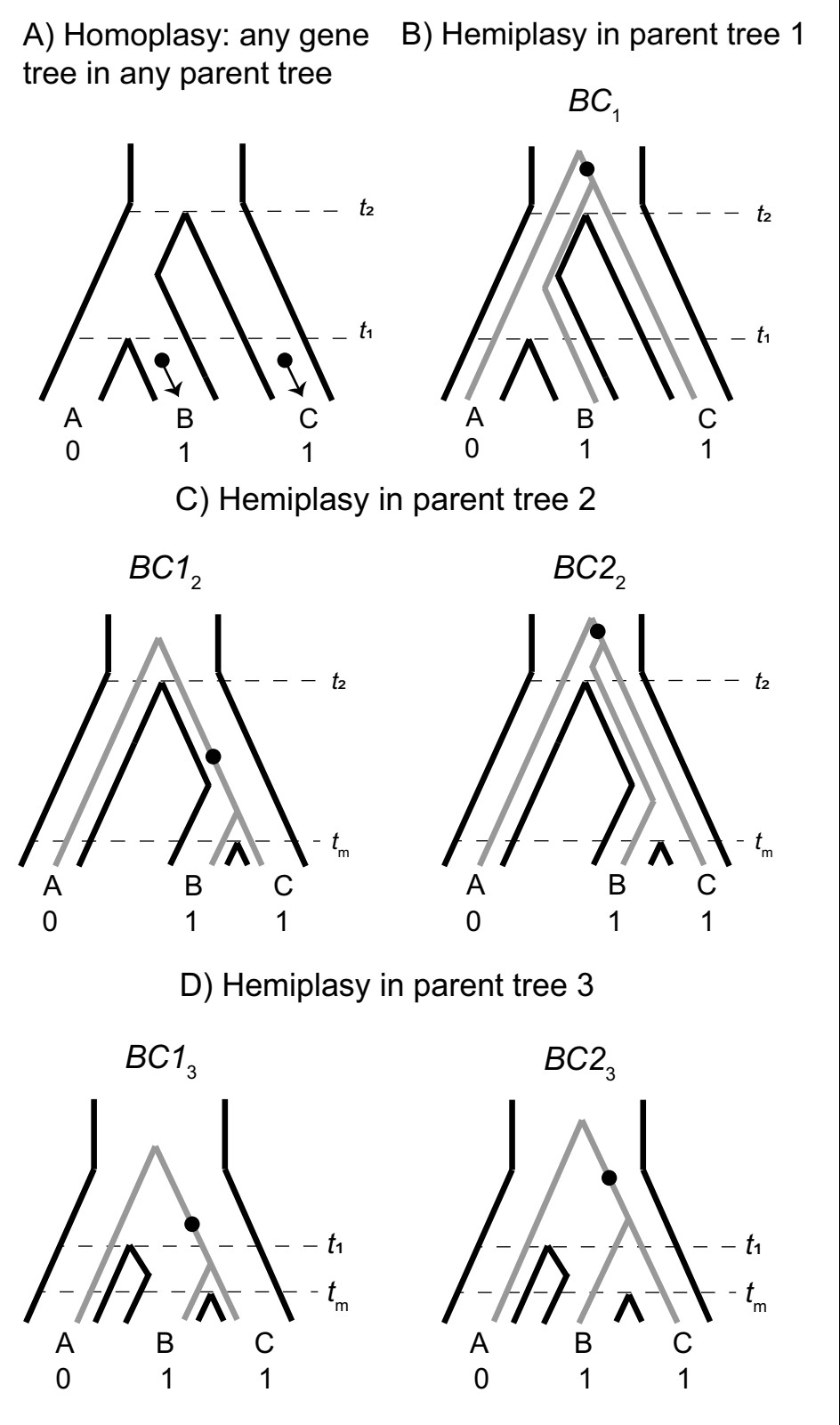

**Figure 3.** The possible paths to homoplasy and hemiplasy under the multispecies network coalescent. Homoplasy can happen on any gene tree, as long as there are two independent mutations on tip branches (panel **A**). Homoplasy can also happen via a mutation in the ancestor of all three species, followed by a reversal (not shown). All cases of hemiplasy require a transition on the internal branch of a gene tree with the topology ((B,C),A). In parent tree 1 (panel **B**), only one such possible gene tree exists (shown in gray; $BC_1$). In both parent trees 2 and 3 (panels **C** and **D** respectively), there are two

*Figure 3 continued on next page*

Figure 3 continued
possible gene trees with this topology. These gene trees differ in internal branch lengths, depending on the parent tree of origin and whether the tree is the result of lineage sorting ($BC1_2$ and $BC1_3$) or incomplete lineage sorting ($BC2_2$ and $BC2_3$) within introgressed histories.

any topology leading to species A, B, and C respectively; $\lambda_4$ denotes the internal branch of any topology, and $\lambda_5$ the branch subtending the root. The notation $\nu(\lambda, \tau)$ represents the probability of a mutation on branch $\lambda_i$ in genealogy $\tau$, where $\tau$ represents any of the four gene trees from any of the three parent trees. The rates of $0 \rightarrow 1$ and $1 \rightarrow 0$ mutations are assumed to be equal, and the rate among lineages is assumed to be constant. Finally, to describe individual genealogies, we use the notation $XY_{i=1,2,3}$, where $X$ and $Y$ denote the sister taxa, and the subscript $i$ denotes the parent-tree of origin. In cases where a tree topology can be produced by either lineage sorting or ILS, a non-subscripted 1 or 2 is used, respectively. Under the ILS-only model, hemiplasy can only occur through a substitution on branch $\lambda_4$ of genealogy $BC_1$ (*Figure 2—figure supplement 1C*, *Figure 3B*). This occurs with the following probability:

$$P_e[BC_1] = \left(\frac{1}{3}e^{-(t_2-t_1)}\right)\nu(\lambda_4, BC_1)\prod_{i \neq 4}(1 - \nu(\lambda_i, BC_1)) \tag{1}$$

(*Guerrero and Hahn, 2018*). *Equation 1* has three components: the probability of observing genealogy $BC_1$, the probability that a mutation happens on the internal branch of that genealogy, and the probability that no other mutations occur. See section 1 of the Appendix for the full expressions for each mutation probability.

Now consider the phylogenetic network described earlier and shown in *Figure 2*. At an introgressed locus, the parent tree topology is ((B,C), A), but could be either parent tree 2 or 3. Within each of these parent trees, there are two possible gene trees that share this topology: one produced by lineage sorting (*Figure 2—figure supplement 1E*, *Figure 3C*) and one produced by ILS where B and C are still the first to coalesce (*Figure 2—figure supplement 1F*, *Figure 3C*). While these trees have the same topology, their expected frequencies and internal branch lengths differ. These quantities also differ depending on the direction of introgression at the locus, that is whether the history follows parent tree 2 or 3.

We first consider the C $\rightarrow$ B direction of introgression, and genealogy $BC1_2$, which is the result of lineage sorting within parent tree 2. This gives:

$$P_e[BC1_2] = \left(1 - e^{-(t_2-t_m)}\right)\nu(\lambda_4, BC1_2)\prod_{i \neq 4}(1 - \nu(\lambda_i, BC1_2)) \tag{2}$$

While *Equation 2* has the same three core components as *Equation 1*, there are several important differences. First, the gene tree probability is the probability of lineage sorting within parent tree 2, which differs from the probability of ILS within parent tree 1. Second, the lower bound of coalescence is $t_m$ rather than $t_1$, resulting in a higher probability of lineage sorting in parent tree 2 as compared to parent tree 1. Third, because B and C coalesce more quickly in this tree, they share a longer internal branch, which means the probability of mutation on that branch is higher (see section 1 of the Appendix).

ILS within parent tree two produces gene tree $BC2_2$, in which B and C are the first to coalesce in the common ancestor of all three species. The probability of hemiplasy in this case is:

$$P_e[BC2_2] = \left(\frac{1}{3}e^{-(t_2-t_m)}\right)\nu(\lambda_4, BC2_2)\prod_{i \neq 4}(1 - \nu(\lambda_i, BC2_2)) \tag{3}$$

In *Equation 3*, the gene tree probability represents ILS in parent tree 2. This probability is lower than its equivalent in parent tree 1, again because $t_m$ is the lower bound for coalescence. Since the upper bound to coalescence is the same ($t_2$), the probability of a mutation on the internal branch of this gene tree is the same as for $BC_1$ (the ILS topology within parent tree 1). To get the overall probability of hemiplasy due to both ILS and introgression when there is gene flow from C $\rightarrow$ B, we weight the probability from each gene tree (*Equations 1-3*) by the admixture proportion, giving the following:

$$P_e[ILS, \text{C} \rightarrow \text{B}] = (1 - \delta_2)P_e[BC_1] + \delta_2(P_e[BC_{1_2}] + P_e[BC_{2_2}]) \tag{4}$$

From *Equation 4*, we can see that introgression will increase the probability of hemiplasy over ILS alone (*Equation 1*) whenever the probability of hemiplasy from parent tree two is higher than from parent tree 1 (i.e. $P_e[BC_{1_2}] + P_e[BC_{2_2}] > P_e[BC_1]$). This is true whenever $t_2 > t_m$ (see section 2 of the Appendix), which is by definition always true in this model.

Finally, we consider the probability of hemiplasy when introgression is in the direction B $\rightarrow$ C (represented by admixture fraction $\delta_3$). As mentioned previously, this direction of introgression results in an upper bound to coalescence of $t_1$ rather than $t_2$. This is the primary difference between the directions of introgression, affecting both the expected gene tree frequencies and mutation probabilities (compare to *Equations 2 and 3*):

$$P_e[BC_{1_3}] = \left(1 - e^{-(t_1 - t_m)}\right) \nu(\lambda_4, BC_{1_3}) \prod_{i \neq 4}(1 - \nu(\lambda_i, BC_{1_3})) \tag{5}$$

and

$$P_e[BC_{2_3}] = \left(\frac{1}{3}e^{-(t_1 - t_m)}\right) \nu(\lambda_4, BC_{2_3}) \prod_{i \neq 4}(1 - \nu(\lambda_i, BC_{2_3})) \tag{6}$$

For the general probability of hemiplasy, including both directions of introgression, we now have:

$$P_e[ILS, \text{C} \rightarrow \text{B}, \text{B} \rightarrow \text{C}] = (1 - (\delta_2 + \delta_3))P_e[BC_1] + \delta_2(P_e[BC_{1_2}] + P_e[BC_{2_2}]) + \\ \delta_3(P_e[BC_{1_3}] + P_e[BC_{2_3}]) \tag{7}$$

Finally, we consider the probability of homoplasy. As described in *Guerrero and Hahn, 2018*, there are two possible paths to homoplasy for a three-taxon tree where taxa B and C carry the derived state. The first is parallel 0 $\rightarrow$ one mutations on branches $\lambda_2$ and $\lambda_3$ (*Figure 3A*), and the second is a 0 $\rightarrow$ one mutation on branch $\lambda_5$ followed by a 1 $\rightarrow$ 0 reversal on branch $\lambda_1$. Both these paths to homoplasy can happen on any possible genealogy, because every topology contains independent tip branches leading to species B and C, as well as an internal branch ancestral to all three species. This gives the following:

$$P_o = \sum_\tau p(\tau) \left[\nu(\lambda_2, \tau)\nu(\lambda_3, \tau) \prod_{i \neq 2,3}(1 - \nu(\lambda_i, \tau)) + \nu(\lambda_5, \tau)\nu(\lambda_1, \tau) \prod_{i \neq 1,5}(1 - \nu(\lambda_i, \tau))\right] \tag{8}$$

where $\tau$ denotes the set of all possible gene trees. (Note that the sum inside *Equation 8* is multiplied by $\frac{1}{p(\tau)}$ in the main text of *Guerrero and Hahn, 2018*. This is a typo in that paper, but the results presented from their model use the correct expression, $p(\tau)$.) This formulation can also be applied to the extended model with introgression, with the understanding that $\tau$ now also includes the gene trees produced by parent trees 2 and 3. Each gene tree used in this summation will have a different set of mutation probabilities, which are detailed in section 1 of the Appendix.

To understand the analytical effect of introgression on the relative risks of hemiplasy and homoplasy, we plotted the ratio $P_e/P_o$ over a realistic range of admixture proportions, timings, and directions (*Figure 4*). The values of $t_1$ and $t_2$ were held constant at 1 and 3.5 coalescent units, respectively, with a population-scaled mutation rate of $\theta = 0.002$. These settings ensured a constant contribution of incomplete lineage sorting to the risk of hemiplasy, leading to a baseline ratio of hemiplasy to homoplasy, $P_e/P_o$, of 0.818 with no introgression. We varied the admixture proportion from 0 to 10%, and the value of $t_m$ from 0.99 (just after the most recent speciation) to 0.01, for three different direction conditions: C $\rightarrow$ B only, B $\rightarrow$ C only, and equal rates in both directions.

## Introgression makes hemiplasy more likely than incomplete lineage sorting alone

Using our model for the probability of hemiplasy and of homoplasy, we examined the ratio $P_e/P_o$ over a range of different introgression scenarios. This ratio summarizes how much more probable hemiplasy is than homoplasy for a given area of parameter space; for example, a value of $P_e/P_o = 2$ means hemiplasy is twice as likely as homoplasy. We find that the probability of hemiplasy relative to

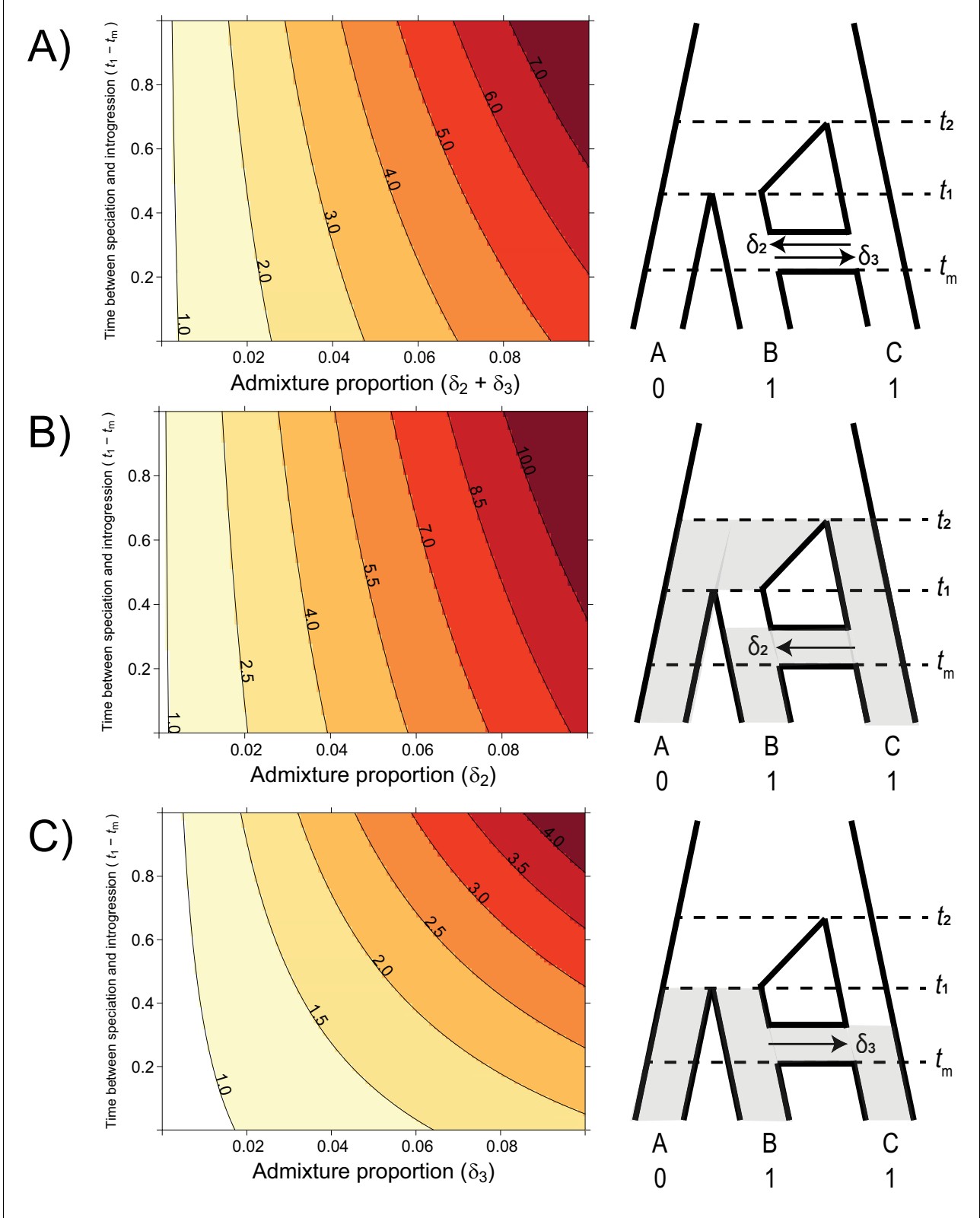

**Figure 4.** The probability of hemiplasy relative to homoplasy (contours) as a function of the admixture proportion (x-axis), the time between speciation and introgression (y-axis), and the direction of introgression (panels). The contours delineate the factor difference between hemiplasy and homoplasy; for instance, a contour value of 2.0 means hemiplasy is twice as probable as homoplasy in that area of parameter space. At x = 0 in each panel, $P_e$/

*Figure 4 continued on next page*

*Figure 4 continued*

$P_o$ = 0.818. (**A**) Equal rates of introgression in both directions. (**B**) Introgression in only the C → B direction. (**C**) Introgression in only the B → C direction.

The online version of this article includes the following source data for figure 4:

**Source data 1.** Data used to generate the contour plots in *Figure 4*.

homoplasy increases as a function of the admixture proportion and how recently introgression occurs relative to speciation (*Figure 4*). As mentioned in the Introduction, there are several possible reasons for these observed trends. The strongest effect on $P_e/P_o$ comes from the admixture proportion: a higher proportion means more loci evolving under parent trees 2 and 3, which means higher frequencies of the genealogies where hemiplasy is possible (i.e. $BC1_2$, $BC2_2$, $BC1_3$, $BC2_3$). The range of simulated admixture proportions from 0 to 10% was meant to capture a biologically plausible range of values, although rates of introgression can sometimes be much higher than this (e.g. *Fontaine et al., 2015*). Even in this modest range, the effect on the probability of hemiplasy can be substantial. We found that an admixture proportion of 5% results in hemiplasy being anywhere from 1.5 to 4 times more likely than homoplasy (depending on the timing and direction of introgression; *Figure 4*). Given the baseline value of $P_e/P_o$ with no introgression for our chosen parameters (0.818), this represents at minimum a doubling of the probability of hemiplasy relative to homoplasy.

The effect of the timing of introgression is more complicated, as it manifests in multiple ways. First, more recent introgression increases the values of $t_2 - t_m$ and $t_1 - t_m$, which in turn increases the degree of lineage sorting in parent trees 2 and 3, respectively. This leads to a higher frequency of gene trees where hemiplasy is possible. Second, the expected length of the internal branches in these two genealogies increases as introgression becomes more recent, which leads to a higher probability of mutations occurring on these branches. Third, since the total height of each tree is being held constant, more recent introgression reduces the lengths of the tip branches leading to species B and C. This reduces the probability of homoplasy due to parallel substitutions, again making hemiplasy relatively more likely. Finally, the strength of the effect of the timing of introgression increases with the admixture proportion, since it is a property of introgressed loci; in other words, the values of of $t_2 - t_m$ and $t_1 - t_m$ do not matter unless loci follow a history of introgression.

The direction of introgression affects the relationship between the admixture proportion, the timing of introgression, and hemiplasy risk (*Figure 4B and C*). While hemiplasy becomes more likely than homoplasy with increased admixture in either direction, $P_e/P_o$ is lower in any given part of parameter space for B → C introgression (*Figure 4C*). This is because the bounds of coalescence for parent tree 3 are $t_1$ and $t_m$, which are always closer in time than $t_2$ and $t_m$ (*Figure 2*). The smaller internal branch in parent tree 3 leads to a higher rate of ILS, in addition to a shorter internal gene tree branch (and lower mutation probability) on genealogies that undergo lineage sorting in these histories. Finally, the timing of introgression has a stronger effect on $P_e/P_o$ in the B → C direction (*Figure 4C*). This is likely because parent tree 3 is truncated relative to parent tree 2 (see *Figure 2*), and so the difference $t_1 - t_m$ makes up a proportionally larger part of the tree height.

### *HeIST*: *H*emiplasy *I*nference *S*imulation *T*ool

As described above, it is possible to infer the most likely number of transitions for an incongruent trait while accounting for discordance in a rooted tree with three taxa. However, similar calculations are computationally difficult for larger numbers of taxa. Here, we present a tool built on top of the coalescent simulator *ms* (*Hudson, 2002*) and sequence simulator *Seq-Gen* (*Rambaut and Grassly, 1997*) that provides an intuitive way to interrogate the parameter space of larger trees. Our tool, dubbed *HeIST*, takes a phylogenetic tree (including an option to specify introgression events) with observed character states as input and returns a simulated distribution of the number of transitions necessary to explain those character states. Introgression events must be specified as an instantaneous 'pulse' from one lineage to another, but we allow flexibility with respect to the timing of that pulse, as well as the rate, direction, and the lineages involved. The input phylogeny must be in coalescent units, but we also include a tool for converting trees given in units of substitutions per site to coalescent units, as long as branches are also associated with concordance factors (see section entitled 'Inferring the tip branch lengths of a phylogeny in coalescent units' below).

*HeIST* uses *ms* to simulate a large number of gene trees from the specified species tree or species network, and then simulates the evolution of a single nucleotide site along each of these gene trees using *Seq-Gen*. Loci where the simulated nucleotide states (transformed into 0/1 characters representing ancestral and derived states) match the character states observed on the species tree are taken as replicate simulations of the evolution of the trait being studied. In these 'focal' cases, *HeIST* counts the number of mutations that occurred along the gene tree in each simulation. It also returns information on the frequency of tip vs. internal branch mutations, transition vs. reversal mutations, the distribution of gene tree topologies, and whether gene trees originate from the species branching history or introgression history. Finally, it returns a summary of how much hemiplasy is likely to contribute to observed character states, using Fitch parsimony (**Fitch, 1971**) to obtain a homoplasy-only baseline for comparison. *HeIST* is implemented in Python 3 and the package/source code are freely available from https://github.com/mhibbins/HeIST.

## *HeIST* effectively captures the effects of ILS and introgression on hemiplasy risk

To evaluate the performance of *HeIST*, we simulated across nine conditions with increasing expected probabilities of hemiplasy, across five different trait mutation rates. The results, shown in **Figure 5**, confirm the theoretical predictions shown in **Figure 4**: the probability of hemiplasy increases as a function of decreasing internal branch length (ILS1-ILS3), increasing probability of introgression (INT1-INT3), and more recent introgression (INT4-INT6). The effect of the timing of introgression is weaker than the effect of the introgression rate, also in line with theoretical expectations. These results held true for both the probability conditional on observing the specified trait pattern (**Figure 5A**) and the raw probability (**Figure 5B**).

While the change in the probability of hemiplasy is broadly consistent with theoretical expectations, the probabilities estimated from *HeIST* consistently underestimated the exact values predicted from theory by a small amount (**Figure 5—figure supplement 2**). We suspect this is due to the occurrence of multiple hits on the same branch of a gene tree, which are not accounted for in our theoretical model. Reversals on branches where hemiplasy can occur would slightly reduce the number of observed hemiplasy cases, leading to the observed underestimation. Consistent with our hypothesis, the mean-normalized mean squared error between simulated and expected values is lower for both lower mutation rates and simulated conditions with a shorter internal branch (**Figure 5—figure supplement 3**). Overall, the mismatch between simulations and theory appears to be negligible for lower, more realistic trait mutation rates, so we do not believe this will be a concern for most empirical applications.

When the parameters of a phylogenetic network are estimated from empirical data, it is possible that many different parameter combinations may be equally likely, especially when only a subset of features are used to fit the model. However, these combinations may differentially affect the probability of hemiplasy: for instance, if the frequency of gene trees is used to fit the network model, but the length of gene tree branches is ignored. To investigate this, we applied *HeIST* to five simulated conditions in which the probability and recency of introgression were increased, while the frequency of the discordant gene tree that could cause hemiplasy was held constant (**Figure 5—figure supplement 4**). We found that, despite a constant gene tree probability, the conditional probability of hemiplasy increased in each successive condition as introgression became more recent and frequent (**Figure 5—figure supplement 5**). These results to some extent merely serve to reinforce the notion that introgression has an effect on hemiplasy above and beyond the effect of ILS alone: by lengthening the branch on the discordant tree that hemiplastic mutations can occur on, introgression has a larger effect than ILS alone. But even when network models that include introgression are used, the estimated effects on hemiplasy will be conservative if parameters are estimated using gene tree frequencies alone.

To evaluate the effects of using *HeIST* on real data, we compared results using the 'true' species tree (**Figure 5—figure supplement 6A**) to those obtained from estimating the species tree with branch lengths using simulated DNA data. This data was run through a pipeline involving estimating a phylogeny using *RAxML*, converting branch lengths to coalescent units, and smoothing (**Figure 5—figure supplement 6B and C**). In all cases, the tree was comprised of eight taxa with no introgression, with three incongruent taxa sharing a hypothetical derived character with a mutation rate of 0.05 per 2*N* generations. Regardless of whether the 'extend' or 'redistribute' method was used for

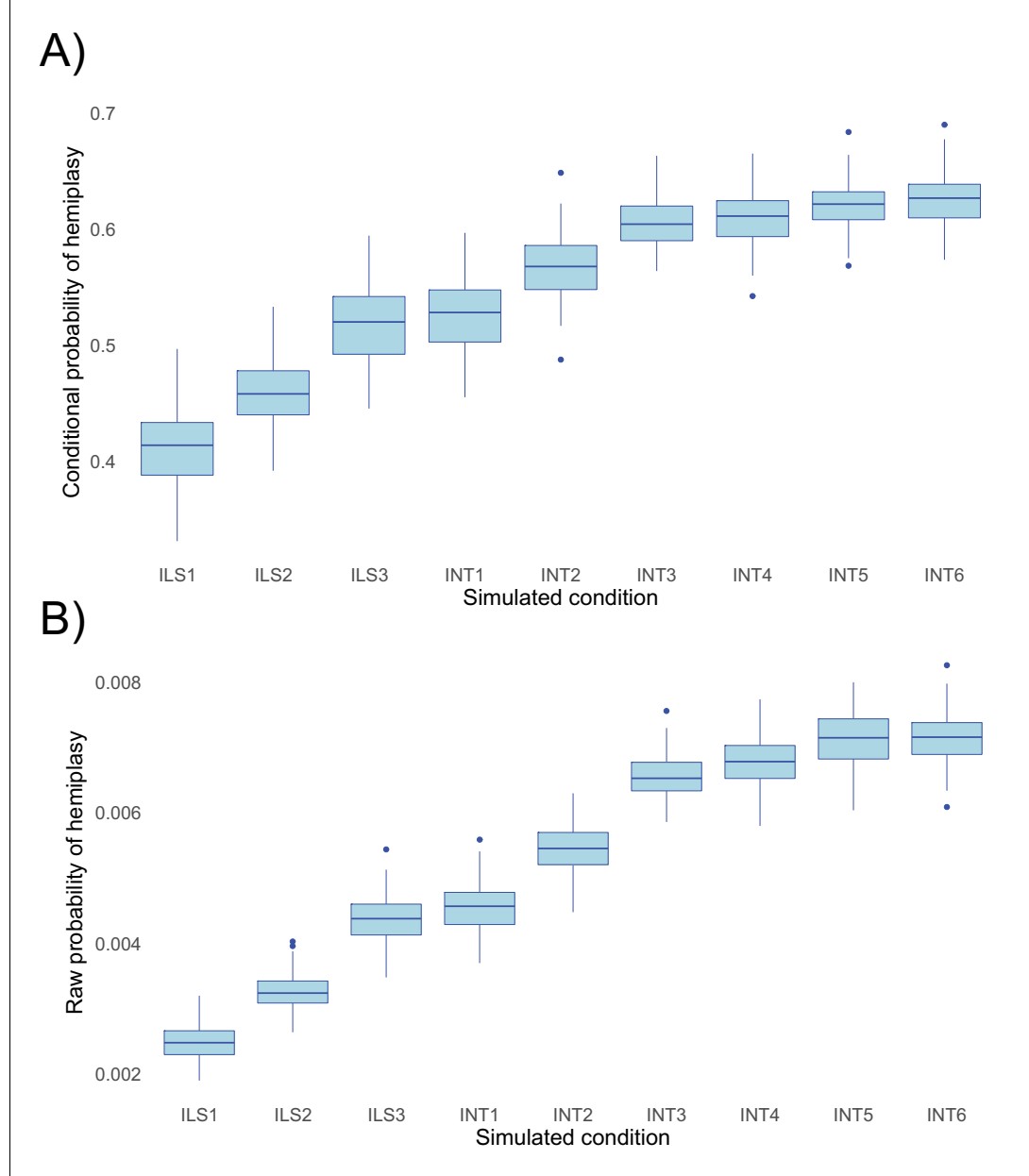

**Figure 5.** Probabilities of hemiplasy estimated from *HeIST* across nine simulated conditions. ILS1-ILS3 decrease the internal branch length of the species tree; INT1-INT3 introduce introgression between derived taxa with increasing probability; INT4-INT6 make introgression more recent while holding the probability constant. See *Figure 5—figure supplement 1* for the exact parameters used in each condition. Panel **A** shows the probability conditional on observing the trait pattern, whereas panel **B** shows the raw probability out of 100,000 simulations.

The online version of this article includes the following source data and figure supplement(s) for figure 5:

**Source data 1.** Data used to generate *Figure 5*.

**Figure supplement 1.** Parameters used for benchmarking simulations in HeIST.

**Figure supplement 2.** Mismatch between simulated (blue boxes) and theoretical (red diamonds) raw probabilities of hemiplasy across simulated conditions, using a mutation rate per 2N generations of 0.05.

**Figure supplement 3.** Degree of mismatch between simulated and theoretical values of the raw probability of hemiplasy for our nine simulated conditions (colors) across five mutation rates (x-axis).

**Figure supplement 4.** Parameters used for simulations demonstrating trade-offs in introgression parameters in HeIST.

**Figure supplement 5.** Trade-offs of different network parameters in HeIST.

**Figure supplement 6.** Effect of phylogenetic inference, branch length unit conversion, and smoothing on estimated probabilities of hemiplasy in HeIST.

smoothing, the overall effect of estimating the tree from sequence data was to lengthen both internal and tip branch lengths, reducing the conditional probability of hemiplasy relative to when the true tree was used (see *Figure 5—figure supplement 6* for exact probabilities). These results suggest that when our unit-conversion approach and smoothing are applied to empirical datasets, the resulting probability estimates will be conservative with respect to the hypothesis of hemiplasy.

## The distribution of green-blooded New Guinea lizards is likely to have arisen from fewer than four transitions

We investigated the most likely number of transitions to green blood from a red-blooded ancestor in New Guinea lizards of the genus *Prasinohaema* (*Rodriguez et al., 2018*). Phylogenies constructed using *RAxML* (*Figure 1*, *Figure 1—figure supplement 1*) and *ASTRAL* (*Figure 1—figure supplement 2*) recover the phylogeny published by *Rodriguez et al., 2018*, including the placement of green-blooded species, and also confirm the existence of very short internal branches. In line with this observation, site concordance factors estimated from UCEs indicate very high rates of discordance in this clade, with some approaching a star tree (i.e. all topologies having frequencies of 33%) (*Figure 1*, *Supplementary file 1*). This strongly suggests that the apparent convergent evolution of the green blood phenotype has been affected by hemiplasy.

We used *HeIST* with the 15-taxon subclade containing six green-blooded species to determine the most likely number of trait transitions. Using our branch length unit conversion tool *subs2coal*, we obtained a best fit line of $y = 0.3038 + 157.03x$ with an adjusted $R^2$ of 0.554 (*Figure 6—figure supplement 1A*). This formula was used to predict the tip branch lengths of the lizard phylogeny in coalescent units, for input to *HeIST* (*Figure 6—figure supplement 2*). This analysis was repeated using two different outgroups, which differed in their distance from the focal subclade. The results were essentially the same using both outgroups; here, we present probabilities using the closer outgroup, *Scincella lateralis*. After simulating $10^{10}$ loci from the lizard phylogeny using *HeIST*, we obtained 2042 loci with a distribution of derived states that matched the empirical distribution of green-blooded species. It is important to note that this number is expected to be a very small proportion of the total number of simulated loci. This occurs because it is necessary to simulate trait histories randomly, but we use only the ones that match the observed distribution. Due to the enormous space of possibilities, the probability of any single trait distribution will be very low, especially with large numbers of taxa and high rates of trait incongruence.

With four independent transitions required without discordance, there are three possible scenarios that involve at least one hemiplastic transition (*Figure 6A*). The first is a hemiplasy-only scenario, in which all green-blooded species are grouped into a single monophyletic clade in a gene tree, and a single transition in the ancestor of this clade explains the observed distribution (*Figure 6A*, left). Out of 2042 focal loci, 726 (35.5%) correspond to this hemiplasy-only case. In the second case, the green-blooded species may be grouped into one or two clades in a gene tree, and there are two independent transitions—at least one of which must involve a discordant ancestral branch (*Figure 6A*, mid-left). Since there are still multiple independent transitions, this case represents a combination of hemiplasy and homoplasy, but exactly which mutations on which branches are hemiplasy vs. homoplasy will depend on the gene tree topology. Of 2042 focal cases, 1316 (64.5), correspond to this scenario. In the third case, the green-blooded species are grouped into as many as three clades, with three independent transitions, at least one of which must be hemiplastic (*Figure 6A*, mid-right). Finally, the green-blooded species may be grouped into four clades, with four independent transitions, as in the species tree (*Figure 6A*, right). We observed no instances of the latter two cases out of 2042 focal loci. These results strongly support the conclusion that, due to hemiplasy, the green-blooded phenotype arose from one or two independent transitions, rather than four.

In all 2042 simulated focal cases, the gene trees on which mutations arose grouped the green-blooded species as monophyletic, regardless of the number of mutations that occurred on the tree. In addition, almost all these monophyletic clades share the same structure, containing two subclades: one containing *P. semoni*, *P. prehensicauda*, *P. flavipes*, and *P. sp nov 1*; another containing *P. virens* and *P. sp nov 2*. It is important to note that the frequency of monophyletic groupings is not expected to reflect the overall distribution of gene trees, but rather the distribution conditional on observing the trait incongruence of interest. These observations make our estimated probabilities

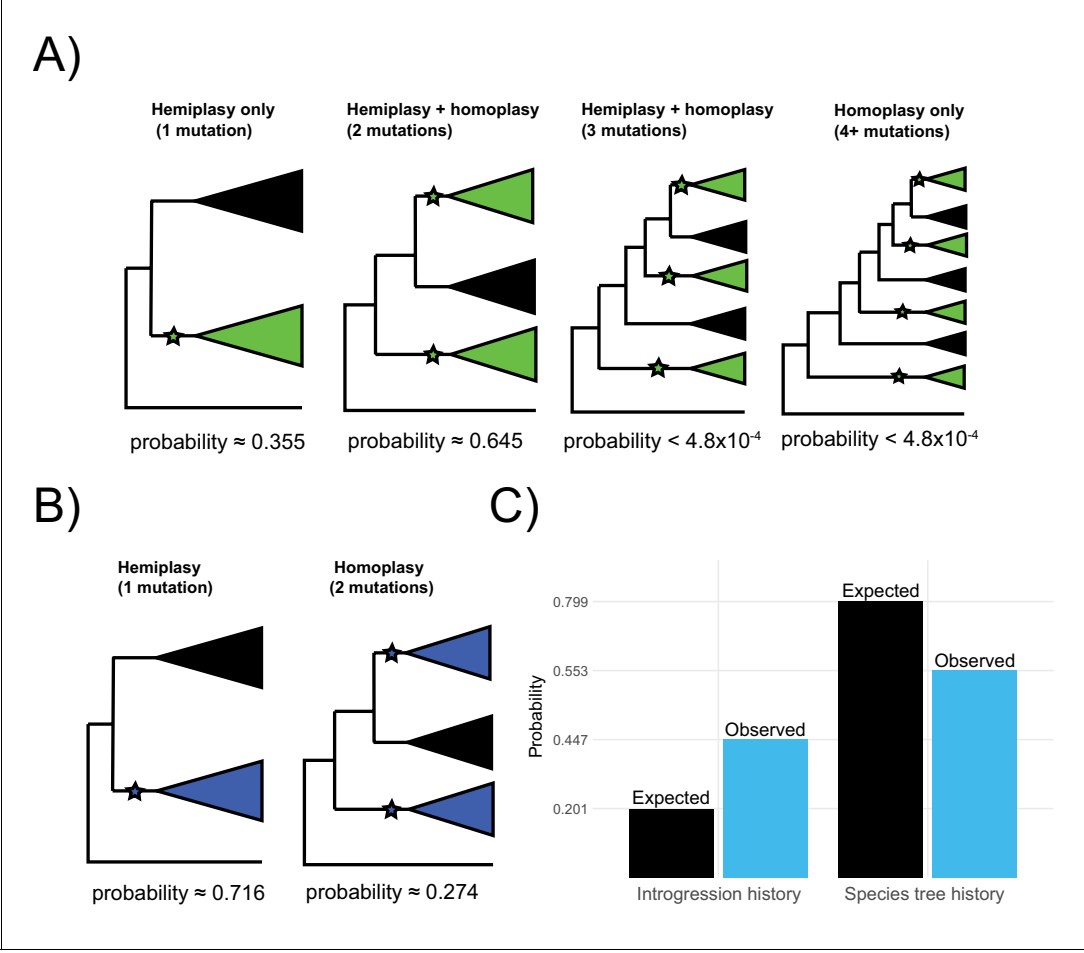

**Figure 6.** Probable histories for (**A**) the origin of green blood in New Guinea lizards and (**B**) the chromosomal inversion spanning the gene *cortex* in *Heliconius*, calculated using *HeIST*. Trees depict the maximum number of clades expected for gene tree topologies under each scenario, with green-blooded clades in green and inversion clades in blue. Branches with proposed ancestral-to-derived transitions are labeled with stars. Exactly which species are sorted into these clades can vary, meaning many possible gene trees exist for each of the depicted scenarios. Correspondingly, any of the labeled hypothetical mutations could represent hemiplasy or homoplasy (except in the case of a single mutation, which must be hemiplasy), depending on the gene tree topology. Reported probabilities are based on $10^{10}$ simulated trees for New Guinea lizards and $10^7$ trees for *Heliconius*, with probabilities conditional on matching the empirical trait distributions. Panel **C** shows what proportion of gene trees originate from a history of introgression vs. the species tree for the results summarized in panel B (blue) as compared to what would be expected based on the inferred network in *Figure 1* (black).

The online version of this article includes the following source data and figure supplement(s) for figure 6:

**Source data 1.** Output file from the *HeIST* lizard analysis, from which the probabilities are reported in *Figure 6A*.

**Source data 2.** Output file from the *HeIST* butterfly analysis, from which the probabilities are reported in *Figure 6B and C*.

**Figure supplement 1.** Regression of internal branch lengths in substitutions per site (x-axis) against the same branch estimated in coalescent units using concordance factors (y-axis) for the 43-species lizard phylogeny (panel **A**) and the 6-species Heliconius phylogeny (panel **B**).

**Figure supplement 2.** Ultrametric phylogeny of green-blooded lizards, with branch lengths in units of 2N generations and green-blooded taxa labeled in green.

easy to interpret: if there was one mutation, it happened in the ancestor of the green-blooded clade; if there were two mutations, they most likely occurred in the ancestors of the two subclades.

Following the logic of 'phyloGWAS' (*Pease et al., 2016*), we checked biallelic sites in the UCE alignment and topologies from the UCE gene trees for a monophyletic clade of green-blooded lizards in order to identify regions potentially associated with variation in blood color. However, both the gene tree and UCE datasets contained missing samples, which made it difficult to confidently identify truly monophyletic clades. On average, approximately nine taxa were unrepresented in the tips of individual gene trees, and approximately 10 were not assigned a base at individual sites in

UCEs. The identity of the missing taxa varied across sites and trees, but often included species inside the 15-taxon subclade containing the green-blooded species, which made it more difficult to consistently polarize and compare patterns of monophyly. In the small proportion of gene trees and UCE sites where information was available for all taxa, we did not find any monophyletic groupings of green-blooded species.

### A chromosomal inversion in the *Heliconius erato/sara* clade likely has a single origin

In addition to the analysis of green-blooded lizards, we also investigated the origins of a chromosomal inversion in the *Heliconius erato/sara* clade (*Edelman et al., 2019*). This inversion spans the gene *cortex*, which is known to influence wing patterning and coloration across butterflies (*Joron et al., 2006*; *Nadeau et al., 2016*). While parsimony applied to the species phylogeny would suggest two independent origins of the inversion (*Figure 1B*), there is clear evidence in *Edelman et al., 2019* of both incomplete lineage sorting and introgression among the clades sharing the inversion, implicating a role for hemiplasy.

We inferred branch lengths in coalescent units for the phylogenetic network of these species given in *Edelman et al., 2019*. Using our unit conversion tool, we obtained a best-fit line of $y = -1.815 + 302.49x$ with an adjusted $R^2$ of 0.98 (though as a note of caution with the $R^2$, this regression contained only five data points; see *Figure 6—figure supplement 1B*). The predicted branch leading to the outgroup was extremely long (~40$N$ generations), so the tree was smoothed using the 'extend' method without the outgroup, and the outgroup was re-added post-smoothing at a length proportional to the original network. The two most highly supported introgression events in the inferred phylogenetic network were then added to the coalescent tree with their previously inferred direction, rate, and approximate timing, before being given to *HeIST* as input (*Figure 1B*).

Using *HeIST*, we found that a single origin of the inversion was most likely, representing 660 of 923 (71.5%) focal cases (*Figure 6B*, left). The scenario involving two mutations was less likely, but was still found in 253 of 923 cases (27.4%) (*Figure 6B*, right). We also observed a small number (3/923, 0.32%) of focal cases with three independent transitions. Overall, our results support the original findings of *Edelman et al., 2019* that the inversion likely arose once and then was shared between lineages via introgression.

Out of 923 simulated loci matching the trait pattern, we found that 413 originated from an introgressed history. This proportion (0.447) is substantially higher than the sum of introgression probabilities specified in the input (0.201), which suggests that introgression contributes more to the probability of observing the trait incongruence than would be expected by chance. In addition, as in the lizard simulations, we found that almost every simulated focal tree (913/923, 98.9%) grouped the *Heliconius* species that share the inversion as monophyletic. However, there is more variation in the structure of the subclades than there was in the lizards. Nevertheless, we can infer from this that two-mutation cases are most likely to arise as independent mutations in the ancestors of two subclades that are part of a larger monophyletic group.

## Discussion

Phenotypic convergence among species can provide important evidence for natural selection. The molecular variation underlying this convergence can arise through independent mutations at the molecular level (*Storz, 2016*). However, it has recently become clear that such cases of 'true' convergence need to be distinguished from cases of apparent convergence due to hemiplasy (*Hahn and Nakhleh, 2016*). Some effort has been made in this regard, through the use of coalescent simulation, summary statistics, and updated comparative approaches (*Pease et al., 2016*; *Copetti et al., 2017*; *Guerrero and Hahn, 2018*; *Wu et al., 2018*). However, these approaches often assume incomplete lineage sorting as the only source of discordance, and cannot explicitly resolve the number of transitions required to explain a trait distribution while accounting for discordance. More recently, *Bastide et al., 2018* and *Karimi et al., 2020* developed extensions to comparative methods that allow quantitative trait likelihoods to be calculated on phylogenetic networks. However, while phylogenetic network inference methods are often robust to the effects of ILS (*Solís-Lemus and Ané, 2016*; *Wen et al., 2018*), the estimated networks themselves do not contain the

necessary information to simultaneously capture the effects of ILS and introgression on trait probabilities (*Mendes et al., 2018*).

Here, we take two important steps toward addressing these problems by: (1) studying the effect of introgression on the risk of hemiplasy under the multispecies network coalescent model and (2) providing a tool that can infer the most probable number of transitions given a phylogenetic distribution of binary traits. We find that introgression increases the risk of hemiplasy over ILS alone, and uncover likely hemiplastic origins for the evolution of green blood from a red-blooded ancestor in New Guinea lizards, and a chromosomal inversion spanning a gene important for wing coloration in *Heliconius*. While our work has important implications for studies of trait evolution, it also carries numerous limitations and simplifying assumptions, which suggest logical next steps for further work. Below we discuss these implications, assumptions, and future directions.

## The probability of hemiplasy due to introgression

A multitude of studies have revealed the potential role of introgression in shaping phenotypic convergence and adaptation (e.g. *Heliconius Genome Consortium, 2012*; *Huerta-Sánchez et al., 2014*; *Jones et al., 2018*; *Mullen et al., 2020*). However, such studies rarely consider how introgression could lead to false inferences of convergence, due to hemiplasy at both the molecular and phenotypic levels, if left unaccounted for. Our model results show that both ILS and introgression must be accounted for in order to make robust inferences of convergent evolution.

Our model for the probability of hemiplasy with introgression, combining concepts from two previously published models, also shares most of their assumptions. First, we have assumed the simplest possible introgression scenario, involving a single pair of species and with introgression occurring instantaneously at some point in the past. However, much more complex introgression scenarios are possible, including introgression between multiple species pairs, involving ancestral populations (and internal branches), at multiple time points in the past, or continuously over a period of time. Horizontal gene transfer, which is more common in prokaryotes, would also require networks that contain reticulation edges spanning very long periods of time. It is not always clear how the probability of hemiplasy would be affected under these alternative introgression scenarios. For example, we assume that the taxa sharing the derived state are also the ones involved in introgression, but introgression between other species pairs could alter patterns of discordance and therefore affect the hemiplasy risk, albeit less directly. Many of these scenarios could be incorporated into the general MSNC framework as additional parent trees, but with more complex histories this may become mathematically intractable even in the three-taxon case; our hemiplasy inference tool, *HeIST*, is designed to ameliorate this issue. Despite these limitations, we can generally expect that introgression will increase the overall risk of hemiplasy whenever rates of introgression are higher between pairs of species that also share the derived state for an incongruent trait. This is because what truly matters is the generation of gene tree topologies with internal branches where hemiplastic transitions can occur; the increased variance in coalescence times under more complex introgression scenarios, while affecting mutation probabilities, should have a comparatively minor effect (*Figure 4*).

We also assume that the coalescence times and gene tree frequencies of loci underlying trait variation follow neutral expectations, even though alleles controlling trait variation are often under some form of selection. Directional selection on such variation will reduce $N_e$ relative to neutral expectations, which will decrease the rate of incomplete lineage sorting and consequently hemiplasy due to ILS. Of course, the amount of ILS used in our simulations is not taken directly from neutral expectations, but rather is estimated from real data. Therefore, the effects of selection on traits of interest will only be manifest if they are greater than the general effects of linked selection across the regions used to estimate discordance (*Kern and Hahn, 2018*). On the other hand, introgressed alleles can lead to hemiplasy even in cases where there is no ILS. In fact, directional selection would also make it more likely that introgressed loci have a discordant topology, as it reduces ILS within parent trees 2 and 3. Alternatively, balancing selection can maintain ancestral polymorphism and increase rates of discordance due to ILS. This will also increase the risk of hemiplasy (e.g. *Fontaine et al., 2015*; *Lamichhaney et al., 2016*; *Palesch et al., 2018*).

## Considerations for the inference tool *HeIST*

While the software we introduce here allows for multiple novel types of inferences, it also has several limitations that are important to address. Errors common to all phylogenetic methods can be introduced into the user-specified species tree/network at several steps, including errors in ortholog identification, tree topology, concordance factors, and branch lengths (via both the conversion to coalescent units and tree smoothing). The process of smoothing the coalescent tree should introduce predictable biases in branch length estimates. When using *ete3*'s method for redistributing branch lengths, internal branches that are very short may have their length increased; conversely, long external branches may be shortened. The lengthening of internal branches decreases the overall rate of discordance, and makes inferences about hemiplasy from *HeIST* conservative. Similarly, when smoothing is done using our function for extending tip branch lengths, the probability of independent mutations on those tip branches (i.e. homoplasy) is increased, again making hemiplasy inferences conservative. The results presented in *Figure 5—figure supplement 6* capture the overall effects of errors in phylogeny estimation, branch length prediction, and smoothing.

Errors in inference may affect our approach to branch length unit conversion in several ways. If concordance factors are underestimates—for instance, due to errors in gene tree reconstruction— then the branch lengths in coalescent units will also be underestimates of their true values. The result would be simulations with more ILS and discordance than actually occurred. In cases where there are concerns about branch length estimates, we suggest running *HeIST* across multiple values; for tip branches, the option exists within *HeIST* to use the lower and upper bounds of the prediction interval in addition to the predictions themselves. In addition, if there are tip branch lengths in the original tree that fall outside the range of internal branch length values, the predicted value of those tip branches in coalescent units may be less reliable, since it requires extrapolation beyond the range of datapoints used to fit the regression. Lastly, we note that *Bastide et al., 2018* propose an approach to estimating coalescent tip branch lengths on a network using the method of least-squares between pairwise genetic distances and network pairwise distances. We expect this approach to have very similar performance to ours, since linear regression is done using least-squares and pairwise genetic distances should be highly correlated with concordance factors.

There are also several practical points to consider when applying *HeIST* to empirical data. When researchers have questions about hemiplasy involving either very large phylogenies or very low mutation rates, only a small number of simulated trees may match the incongruent pattern found in real data. The large number of simulations required may not be computationally feasible, although careful pruning of species that do not affect inferences of hemiplasy may greatly reduce this limitation. By default, *HeIST* will prune the input phylogeny to include the smallest subclade that contains all the taxa with the derived state, plus a specified outgroup. In addition, while *HeIST* can simulate phylogenies with introgression, it requires that the timing, direction, and rate of each introgression event is provided. To obtain this information, we recommend using a phylogenetic network-based approach such as PhyloNet (*Wen et al., 2018*), SNaQ (*Solís-Lemus and Ané, 2016*), or the Species-Network (*Zhang et al., 2018a*) package within BEAST2 (*Bouckaert et al., 2019*).

Finally, an issue that concerns both our theoretical work and *HeIST* is the specification of the mutation rate. In both cases, we assume that the rates of $0 \rightarrow 1$ and $1 \rightarrow 0$ transitions are equivalent, and that these rates are constant across the tree under study. Violations of these assumptions will certainly influence the probabilities of hemiplasy and homoplasy, although it is unlikely that underlying mutation rates will vary substantially among closely related lineages (*Lynch, 2010*). More importantly, these rates represent the mutation rate among character states, and may not always be the same as nucleotide mutation rates. We have assumed in the results presented here that transitions between character states are controlled by a single site, and therefore that the nucleotide mutation rate is a good approximation of the trait mutation rate. However, the degree to which this is true will depend on the genetic architecture underlying a trait. For example, transitions in floral color are often underlain by loss-of-function mutations, and many mutational targets can potentially lead to the same phenotypic changes (*Rausher, 2008*; *Smith and Rausher, 2011*). In such cases, the rate of trait transitions can potentially be many times higher than the nucleotide mutation rate, with homoplasy becoming more probable as a result. In contrast, trait transitions can also require multiple molecular changes, the order of which may be constrained by pleiotropy and epistasis. Such changes underlie, for instance, high-altitude adaptation of hemoglobin in mammals (*Storz et al., 2009*;

*Tufts et al., 2015*). In these cases, the rate of trait transitions may be many times lower than the nucleotide mutation rate, with hemiplasy becoming more probable as a result.

## Evolution of green-blooded lizards and the H*eliconious* inversion

In our analysis of lizards in the genus *Prasinohaema*, we found strong support for one or two independent origins of green blood from a red-blooded ancestor, with two origins being the most likely. This contrasts with analyses that do not account for gene tree discordance, in which four transitions is the best explanation. In *Heliconius*, we found support for a single origin of a chromosomal inversion, in contrast to methods that do not account for discordance. Both these results strongly suggest that hemiplasy has played a role in the evolution of these traits.

Applications of *HeIST* to these clades involves some system-specific assumptions, the first of which relates to the genetic architecture of the traits under study. For the lizard analysis, it invokes the potentially strong assumption that the green-blooded phenotype is achievable by a single muta-tion. While the physiological mechanism for this phenotype is well-understood (*Austin and Jessing, 1994*), the genetic architecture underlying the transition from a red-blooded ancestor is not. As discussed in the previous section, this architecture will affect the choice of $\theta$ used as the trait evolution-ary rate in our simulations. Since the genetic architecture is unknown, our choice of $\theta$ was based on what is typically observed for nucleotide mutations in vertebrate systems (*Lynch, 2010*). For the *Heli-conius* inversion, the architecture is more clear-cut, since chromosomal inversions are a single muta-tional event by definition. While the per-generation rate of de novo chromosomal inversions is not known for many systems, it is certain to be lower than the rate for nucleotide mutations per-site. Nucleotide $\theta$ is estimated at 0.02–0.03 for *H. melpomene* (*Martin et al., 2016*), and averages around 0.01 in invertebrates (*Lynch, 2010*). Our choice of $\theta$ for the inversion was one order of magnitude lower than these estimates.

Another key assumption is that the estimated gene trees and concordance factors are accurate, as is the regression approach for converting branch length units. The observed $R^2$ of 0.554 for the unit-conversion in the lizard dataset might be interpreted as surprisingly low given that it is a regres-sion of the same quantity measured in two different units. This value likely reflects uncertainty gener-ated in several steps of our analysis, including the estimation of branch lengths in the maximum-likelihood species tree, and the procedure of randomly sampling quartets to estimate sCFs used by *IQ-TREE*. In *Heliconius*, the $R^2$ was much higher at 0.98, but with only five data points there was lim-ited information about the true relationship. Nonetheless, we observed the expected positive corre-lation in both cases, and a sufficient amount of variation is explained to ensure that tip branches estimated in coalescent units are proportionally similar to those in the maximum-likelihood tree, sug-gesting that the regression approach works well as an approximation. In addition, the regression line on the lizard data appears to slightly over-estimate very short branch lengths in coalescent units, making our inferences of hemiplasy conservative.

## Conclusions

A major question in the study of convergent evolution is whether phenotypic convergence is under-lain by convergent changes at the molecular level (*Storz, 2016*). The work presented here is con-cerned primarily with such molecular changes, and the results of our empirical analyses highlight how apparently convergent phenotypes can arise from a single molecular change. Such shared changes come about as a result of gene tree discordance due to ILS, introgression, or some combi-nation of the two. Given that these phenomena are common in phylogenomic datasets (*Pollard et al., 2006*; *Fontaine et al., 2015*; *Pease et al., 2016*; *Novikova et al., 2016*; *Wu et al., 2018*), perhaps it should be less surprising that phylogenetically incongruent traits often have a com-mon genetic basis.

Finally, while the tools presented here may help to rule out cases of molecular convergence, the observation of a single molecular origin for a trait does not rule out the occurrence of convergent adaptation in general. Parallel selective pressures from the environment on the same molecular vari-ation may be regarded as one of many possible modes of convergent evolution (*Lee and Coop, 2017*). In studying novel phenotypes such as green blood or wing patterning and coloration, there is still tremendous interest in understanding the ecological pressures that may have led to the indepen-dent fixation of single, ancestral changes along multiple lineages. In general, integrative approaches

combining modern phylogenomics with an ecological context will pave the way toward an improved understanding of the nature of convergent evolution.

## Materials and methods

### Accuracy of *HeIST*

To confirm that *HeIST* accurately counts mutation events, and is consistent with our theoretical findings, we evaluated its performance under nine simulated conditions with increasing levels of expected hemiplasy. All simulated conditions involve a four-taxon tree with the topology (((4,3),2),1). Species 4 and 2 carry the derived state for a hypothetical binary character. The split of species one from the ancestor of 4, 3, and 2 occurs at $8N$ generations in the past. The first three simulated conditions contain no introgression, and progressively decrease the length of the internal branch subtending species 4 and 3. The total tree height was held constant. The simulated internal branch lengths were $2N$, $1.5N$, and $N$ generations for conditions *ILS1*, *ILS2*, and *ILS3* respectively. The subsequent six conditions maintain the *ILS3* condition for branch lengths, with the addition of an introgression event from species 2 into species 4. For conditions *INT1*, *INT2*, and *INT3*, the timing of introgression was held constant at $0.6N$ generations, while the introgression probability was set to 0.01, 0.05, and 0.1, respectively. For conditions *INT4*, *INT5*, and *INT6*, the introgression probability was held constant at 0.1, while the timing of introgression was reduced to $0.4N$, $0.2N$, and $0.1N$, respectively. The parameters used for each condition are summarized in *Figure 5—figure supplement 1*.

We performed two sets of simulations: (1) 100 replicates of each condition, consisting of 100,000 gene trees each, with a constant mutation rate of 0.05 per $2N$ generations; (2) 20 replicates of each condition, for each of five different mutation rates per $2N$ generations (0.0005, 0.0025, 0.005, 0.025, 0.05), each consisting of 1,000,000 gene trees. For each combination of parameters, we estimated the probability of hemiplasy conditional on observing the specified trait pattern, and the raw probability of hemiplasy out of the total number of replicates. For the latter simulation set, we estimated the mean-squared error (MSE) using the simulated values as observations and the expected value from theory as the true mean. These MSE values were divided by the simulated mean to compare error across conditions with different ranges of expected values.

Trade-offs among parameters mean that many combinations of estimated network parameters may be equally consistent with patterns in subsets of the observed data. To investigate possible effects on the probability of hemiplasy, we evaluated the performance of *HeIST* under five additional simulation conditions (*Figure 5—figure supplement 4*). In each successive condition, the probability of introgression was increased, while the timing of introgression was made more recent. The length of the internal branch in the species tree was also increased such that the expected frequency of the discordant gene tree that causes hemiplasy remained approximately constant (*Figure 5—figure supplement 5A*). These simulations used the same tree topology, derived taxa, split time of the ancestral population, and mutation rate as the first set of benchmarking simulations. Condition 1 used the same parameters as *ILS1*. Conditions 2–5 used the following sets of parameters, respectively: $2.08N$, $2.32N$, $2.6N$, $2.8N$ generations for the length of the internal branch; 0.01, 0.025, 0.04, 0.05 for the probability of introgression; $0.4N$, $0.3N$, $0.2N$, $0.1N$ generations for the timing of introgression (*Figure 5—figure supplement 4*). For each condition, we performed 100 replicate simulations of 100,000 gene trees each in *HeIST*, and estimated the probability of hemiplasy conditional on observing the trait pattern.

### Inferring the tip branch lengths of a phylogeny in coalescent units

Inferences made under the multispecies coalescent require branch lengths specified in coalescent units. However, most standard methods for building phylogenies infer branches in units of substitutions per site. Units of absolute time inferred from substitution rates using molecular clock approaches can be converted into coalescent units, provided that the generation time and effective population size are known. However, these parameters are sometimes not available or accurate for a given system. As an alternative, estimates of gene tree discordance can be used to estimate internal branch lengths in coalescent units, but these provide no information about the lengths of tip branches. For example, the species tree inference software *ASTRAL* (*Zhang et al., 2018b*) does not infer tip branch lengths, while the software *MP-EST* (*Liu et al., 2010*) adds branches of length nine

for every tip. These tip lengths are necessary to make accurate inferences about hemiplasy and homoplasy from empirical data, since they affect the probability of mutation on tip branches.

To ameliorate this problem, we have applied a simple regression approach for inferring tip lengths in coalescent units (see *Bastide et al., 2018* for an alternative method). Our approach makes use of concordance factors: estimates of the fraction of concordant loci with respect to a particular branch in a species tree. Concordance factors come in two flavors: gene concordance factors (gCFs) (*Gadagkar et al., 2005*; *Ané et al., 2007*), which estimate the concordance of gene tree topologies, and site concordance factors (sCFs) (*Minh et al., 2020a*), which do the same for parsimony-informative sites. In general, concordance factors estimated from quartets provide an estimate of $1 - \frac{2}{3}e^{-T}$, where $T$ is the length of the internal branch in coalescent units. With concordance factors given on the internal branches of a tree that has lengths in substitutions per site, the aforementioned formula can be used to obtain estimates of those same branch lengths in units of 2$N$ generations. A regression of the internal branch length estimates in both units can then be used to obtain a formula for unit conversion between them. *HeIST* uses this formula to predict the tip branch lengths of the tree in coalescent units. To partially account for uncertainty introduced during tip branch length prediction, *HeIST* can also be run using the lower or upper bounds of the prediction 95% confidence interval as the inferred tip lengths, in addition to the predictions themselves. As a final step in this process, the tree in coalescent units is smoothed, as *ms* requires the input tree to be ultrametric. *HeIST* has two options for how to perform this smoothing. The first redistributes the tree branch lengths so that the distance from the root to each tip is the same; this is done using the convert_to_ultrametric() function in the Python library *ete3* (*Huerta-Cepas et al., 2016*). The second extends the lengths of tip branches while preserving internal branch lengths; this function is coded within *HeIST*, but was borrowed from a commented block in *ete3*'s source code.

To investigate the potential bias introduced to results from *HeIST* by either phylogenetic inference, the branch regression approach, or subsequent smoothing, we compared the outputs of *HeIST* run from an eight-taxon test tree with known branch lengths. To generate realistic datasets, we first simulated 3000 gene trees from the known species tree using *ms*, and then simulated 1 Kb of sequence from each locus with $\theta$ = 0.001 using *Seq-Gen*. These loci were concatenated into a single 3 Mb alignment, which was given to *RAxML* version 8.2.12 (*Stamatakis, 2014*) using the GTR substitution model with rate heterogeneity to infer a species tree in units of substitutions per site. This inferred tree and the concatenated alignment were given to *IQ-TREE* version 2.0 (*Minh et al., 2020b*) to infer site concordance factors. This substitution tree with nodes labeled with concordance factors was given as input to *HeIST*, where our branch regression approach was applied. We tested both methods for tree smoothing from this input. We estimated the probability of each number of mutations conditional on observing the incongruent site pattern, and results from this analysis were compared to those obtained using the 'true' test tree (*Figure 5—figure supplement 6*).

Our regression approach is implemented in *HeIST* and can be run as part of the overall hemiplasy analysis, or separately using the module '*subs2coal*'.

## Empirical applications of *HeIST*

We applied *HeIST* to two empirical case studies where hemiplasy appeared to be a plausible explanation for observed trait incongruences. The first is a dataset of New Guinea lizards (*Rodriguez et al., 2018*). As described in the Introduction, the genus *Prasinohaema* contains six species that have evolved green blood from a red-blooded ancestor (*Figure 1A*). Previous analyses of the species tree built from thousands of loci inferred that four independent transitions are necessary to explain the phylogenetic distribution of green-blooded species (*Rodriguez et al., 2018*). This conclusion is the same using any standard phylogenetic comparative method, whether ancestral state reconstruction is carried out using maximum likelihood (*Rodriguez et al., 2018*) or Fitch parsimony (this study). However, the phylogeny for this clade contains many short branches (*Figure 1*), suggesting that a scenario involving at least some hemiplasy (in this case, 1–3 mutations) may be preferred over homoplasy-only scenarios when discordance is accounted for.

To address this question, we used the original dataset of *Rodriguez et al., 2018*, consisting of 3220 ultra-conserved elements (UCEs) totalling approximately 1.3 Mb for 43 species. We then down-sampled these species to 15 taxa in the clade including the green-blooded species and an outgroup (*Figure 1A*). We constructed a concatenated maximum likelihood species tree, in addition

to gene trees for each UCE, using *RAxML* version 8.2.12 (*Stamatakis, 2014*). To verify the species tree topology for the 15-taxon subclade, we also constructed a tree with *ASTRAL-III* version 5.6.3 (*Zhang et al., 2018a*). Site and gene concordance factors were calculated for this tree using *IQ-TREE* version 2.0 (*Minh et al., 2020a*; *Minh et al., 2020b*). To obtain the phylogeny in coalescent units, we employed the regression approach described above for unit conversion as implemented in *HeIST*. The 'extend' method was used for tree smoothing. We then used *HeIST* to simulate $10^{10}$ loci from the lizard subclade containing green-blooded species, with a population-scaled mutation rate ($\theta$) of 0.0005 per 2*N* generations. While specific parameter estimates are not available for this system, our choice of $\theta$ reflects broad estimates of $N_e$ and μ on the order of $10^5$–$10^6$ (*Lynch, 2006*) and $10^{-8}$ – $10^{-9}$ per-base per-generation (*Lynch, 2010*), respectively, in vertebrates (see Discussion). This analysis was performed for each of two outgroups: *Lygosoma sp*, which is sister to 40 species in the 43-species phylogeny, and *Scincella lateralis*, which is sister to the 15-taxon clade containing the green-blooded species. We also calculated *D*-statistics (*Green et al., 2010*) for 12 trios involving green-blooded taxa, finding no strong evidence of introgression (block bootstrap significance tests, *Supplementary file 1*-Tables 2 and 3). Therefore, our simulations did not include any introgression events.

The second empirical case study involves the origins of a chromosomal inversion spanning a gene important for wing coloration in *Heliconius* butterflies (*Edelman et al., 2019*). The derived inversion arrangement is shared by four taxa, grouped into two subclades in the *erato/sara* group of *Heliconius*. Fitch parsimony suggests two independent origins, but a combination of short internal branches and introgression between the ancestral populations sharing the inversion (*Edelman et al., 2019*) strongly suggests a role for hemiplasy. We obtained the phylogenetic network—that is, the species tree with reticulation edges—inferred in units of substitutions per site, in addition to gene concordance factors, from the authors. As our regression approach for conversion to coalescent units cannot be used on phylogenetic networks directly, we used the species tree embedded in the network with concordance factors as input to *subs2coal*. The two most strongly supported introgression events were then added back onto the smoothed network in coalescent units, using the inferred rates and directions, with approximate timings based on the location of the events in the original network and our requirement that these events be instantaneous 'pulses.' From this input (shown in *Figure 1B*), we simulated $10^7$ gene trees in *HeIST* using a mutation rate of 0.0005 per 2*N* generations. That our choice of $\theta$ for this system is the same as in our lizard analysis is just a coincidence: it reflects a trade-off between the generally higher effective population size for insects (*Lynch, 2006*) and the lower mutation rate expected for chromosomal inversions; see Discussion. We also performed the same simulations without specifying the introgression events to obtain an ILS-only estimate of the probability of hemiplasy.

## Acknowledgements

We thank Rafael Guerrero, Leonie Moyle, and Ben Fulton for helpful comments and advice, as well as three anonymous reviewers, the associate editor, and Chris Austin for suggestions that helped improve this work. We also thank Zachary Rodriguez for sharing the lizard data and Nate Edelman for sharing the *Heliconius* phylogenetic network. This work was supported by National Science Foundation grant DEB-1936187.

## Additional information

### Funding

| Funder | Grant reference number | Author |
|---|---|---|
| National Science Foundation | DEB-1936187 | Matthew W Hahn |

The funders had no role in study design, data collection and interpretation, or the decision to submit the work for publication.

## Author contributions
Mark S Hibbins, Conceptualization, Data curation, Software, Formal analysis, Validation, Investigation, Visualization, Methodology, Writing - original draft, Project administration, Writing - review and editing; Matthew JS Gibson, Data curation, Software, Validation, Methodology, Writing - original draft, Writing - review and editing; Matthew W Hahn, Conceptualization, Supervision, Funding acquisition, Validation, Writing - original draft, Project administration, Writing - review and editing

## Author ORCIDs
Mark S Hibbins (iD) https://orcid.org/0000-0002-4651-3704
Matthew JS Gibson (iD) https://orcid.org/0000-0001-7855-1628
Matthew W Hahn (iD) http://orcid.org/0000-0002-5731-8808

## Decision letter and Author response
Decision letter https://doi.org/10.7554/eLife.63753.sa1
Author response https://doi.org/10.7554/eLife.63753.sa2

# Additional files

## Supplementary files
• Supplementary file 1. Table 1: Site concordance factors from IQtree. 'ID': the ID of the internal-branch in the full lizard phylogeny, as labeled in *Figure 1 —figure supplement 1*. 'sCF': the value of the site concordance factor for the branch, averaged over 100 randomly sampled quartets.'sDF1' and'sDF2': the site discordance factors for the first and second most common discordant site patterns at each branch, respectively.'sN': the average number of informative sites averaged across the sampled quartets at each branch.'Length' is the length of the internal branch in substitutions per site. -Table 2: Trios involving green-blooded species which were evaluated for evidence of introgression using D statistics. Species are listed in the order P1, P2, P3, where an excess of shared P1/P3 or P2/P3 alleles would indicate evidence of introgression. Lygosoma sp was used as the outgroup for each trio. Table 3: D statistic results for the 12 trios listed in Table 1, estimated from the concatenated alignment of ultra-conserved elements (UCEs). Significance was evaluated for each trio by bootstrap-sampling the UCEs to generate a null distribution of alignments, and asking how often the bootstrap distribution of 1000 D statistics was at least as extreme as the observed value.

• Transparent reporting form

## Data availability
Availability of the lizard genomic data and Heliconius phylogenetic network is detailed in the Acknowledgements section of the source manuscript. Source code and test cases for our software HeIST are freely available from the GitHub repository. Source data files have been provided for Figures 1, 4, 5, and 6. The Appendix details all the mutation rate parameters of our mathematical model.

The following previously published datasets were used:

| Author(s) | Year | Dataset title | Dataset URL | Database and Identifier |
|---|---|---|---|---|
| Rodriguez ZB, Perkins SL, Austin CC | 2018 | Raw sequencing reads from ultraconserved elements of Australasian skinks | https://www.ncbi.nlm.nih.gov/bioproject/?term=PRJNA420910 | NCBI BioProject, PRJNA420910 |
| Edelman NB, Frandsen PF, Miyagi M, Clavijo B, Davey J, Dikow RB, Van Belleghem SM, Patterson N, Neafsey DE, Richard C, Kumar S, Moreira GRP, Salazar C, | 2019 | Data from: Genomic architecture and introgression shape a butterfly radiation | http://dx.doi.org/10.5061/dryad.b7bj832 | Dryad Digital Repository, 10.5061/dryad.b7bj832 |

Chouteau M,
Counterman B,
Papa R, Blaxter M,
Reed RD,
Dasmahapatra KK,
Kronforst M, Joron
M, Jiggins CD,
McMillan WO,
Di Palma F,
Blumberg AJ,
Wakeley J, Jaffe D,
Mallet J

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

## Appendix 1

### 1 Mutation probabilities on genealogies

Each of the twelve possible genealogies under our parent tree model has a set of five branch lengths along which mutations can occur. $\lambda_1$, $\lambda_2$, and $\lambda_3$ denote the tip branches leading to species A, B, and C respectively; $\lambda_4$ denotes the internal branch, and $\lambda_5$ denotes the ancestral branch. As described in the supplement of *Guerrero and Hahn, 2018*, the mutation probability on each of these branches has the general form $\int 1 - e^{-\mu x} f(x) dx$, where $\mu$ is the mutation probability per $2N$ generations, x is the random variable for the branch length, and f(x) is the probability density function for x. We begin with the mutation probabilities for parent tree 1, which are found in the supplement of Guerrero and Hahn, and will be re-written here to be consistent with notation. In the following notation, parent tree 1 will be denoted as "pt1". Since many of the genealogies are identical in length, the mutation probabilities on their branches can be written with general expressions. We first consider the genealogies $AB2_1$, $BC_1$, and $AC_1$, which are all produced via incomplete lineage sorting in parent tree 1, and share the following set of mutation probabilities:

$$\nu_1[ILS, pt1] = \frac{1}{\Lambda} \int_0^{t_3 - t_2} (1 - e^{-\mu(t_1 + (t_2 - t_1) + x)}) \frac{3}{2} (e^{-x} - e^{-3x}) dx \tag{1}$$

$$\nu_2[ILS, pt1] = \frac{1}{\Lambda} \int_0^{t_3 - t_2} (1 - e^{-\mu(t_1 + (t_2 - t_1) + x)}) 3e^{-3x} (1 - e^{-((t_3 - t_2) - x)}) dx \tag{2}$$

$$\nu_4[ILS, pt1] = \frac{1}{\Lambda} \int_0^{t_3 - t_2} \int_0^{t_3 - t_2} 3e^{-3y} (\int_0^{(t_3 - t_2) - y} e^{-x} (1 - e^{-\mu x}) dx) dy \tag{3}$$

$$\nu_5[ILS, pt1] = \frac{1}{\Lambda} \int_0^{t_3 - t_2} (1 - e^{-\mu((t_3 - t_2) - x)}) 3e^{-3x} dx \tag{4}$$

In each of the above, $\Lambda = 1 + \frac{1}{2} e^{-3(t_3 - t_2)} - \frac{3}{2} e^{-(t_3 - t_2)}$ is the probability of coalescence of A, B, and C in their ancestral population. $t_3$ denotes the total height of the tree, i.e. the time at the base of the tree. The difference between $t_3$ and $t_2$ determines the duration of the ancestral population of all three taxa, before speciation occurs. *Equations 1* through four each represent the mutation probabilities for multiple branches, which are as follows:

$$\nu_1[ILS, pt1] = \nu(\lambda_3, AB2_1) = \nu(\lambda_1, BC_1) = \nu(\lambda_2, AC_1) \tag{5}$$

$$\begin{aligned} \nu_2[ILS, pt1] &= \nu(\lambda_1, AB2_1) = \nu(\lambda_2, AB2_1) = \\ \nu(\lambda_2, BC_1) &= \nu(\lambda_3, BC_1) = \nu(\lambda_1, AC_1) = \nu(\lambda_3, AC_1) \end{aligned} \tag{6}$$

$$\nu_4[ILS, pt1] = \nu(\lambda_4, AB2_1) = \nu(\lambda_4, BC_1) = \nu(\lambda_4, AC_1) \tag{7}$$

$$\nu_5[ILS, pt1] = \nu(\lambda_5, AB2_1) = \nu(\lambda_5, BC_1) = \nu(\lambda_5, AC_1) \tag{8}$$

The gene tree produced by lineage sorting in parent tree 1, $AB1_1$, has a different set of mutation probabilities, since the branches have different expected lengths. These are:

$$\nu(\lambda_1, AB1_1) = \nu(\lambda_2, AB1_1) = \frac{1}{1 - e^{-(t_2 - t_1)}} \int_0^{t_2 - t_1} (1 - e^{-\mu(t_1 + x)}) e^{-x} dx \tag{9}$$

$$\nu(\lambda_3, AB1_1) = \frac{1}{1 - e^{-(t_3 - t_2)}} \int_0^{t_3 - t_2} (1 - e^{-\mu(t_1 + (t_2 - t_1) + x)}) e^{-x} dx \tag{10}$$

$$\nu(\lambda_4, AB1_1) = \int_0^{t_2 - t_1} \frac{e^{-y}}{1 - e^{-(t_2 - t_1)}} (\int_0^{t_3 - t_2} (1 - e^{-\mu((t_2 - t_1) - y + x)}) \frac{e^{-x}}{1 - e^{-(t_3 - t_2)}} dx) dy \tag{11}$$

$$\nu(\lambda_5, AB1_1) = \int_0^{t_2-t_1} \frac{1}{1-e^{-(t_3-t_2)}} \int_0^{t_3-t_2} (1 - e^{-\mu((t_3-t_2)-x)}) e^{-x} dx \tag{12}$$

Now we consider introgression, starting with parent tree 2. Many of the mutation probabilities are symmetrical with parent tree 1 and therefore remain the same, and the remainder have the same general form with different parameters. For the ILS genealogies $BC2_2$, $AB_2$, and $AC_2$, *Equations 1 and 2* have the time of A-B speciation ($t_1$) replaced with the timing of B-C introgression ($t_m$). This gives:

$$\nu_1[ILS, pt2] = \frac{1}{\Lambda} \int_0^{t_3-t_2} (1 - e^{-\mu(t_m+(t_2-t_m)+x)}) \frac{3}{2}(e^{-x} - e^{-3x}) dx \tag{13}$$

$$\nu_2[ILS, pt2] = \frac{1}{\Lambda} \int_0^{t_3-t_2} (1 - e^{-\mu(t_m+(t_2-t_m)+x)}) 3e^{-3x}(1 - e^{-((t_3-t_2)-x)}) dx \tag{14}$$

$$\nu_4[ILS, pt2] = \nu_4[ILS, pt1] \tag{15}$$

$$\nu_5[ILS, pt2] = \nu_5[ILS, pt1] \tag{16}$$

These correspond to the following branch mutation probabilities:

$$\nu_1[ILS, pt2] = \nu(\lambda_1, BC2_2) = \nu(\lambda_3, AB_2) = \nu(\lambda_2, AC_2) \tag{17}$$

$$\begin{aligned} \nu_2[ILS, pt2] &= \nu(\lambda_2, BC2_2) = \nu(\lambda_3, BC2_2) = \\ \nu(\lambda_1, AB_2) &= \nu(\lambda_2, AB_2) = \nu(\lambda_1, AC_2) = \nu(\lambda_3, AC_2) \end{aligned} \tag{18}$$

$$\nu_4[ILS, pt2] = \nu(\lambda_4, BC2_2) = \nu(\lambda_4, AB_2) = \nu(\lambda_4, AC_2) \tag{19}$$

$$\nu_5[ILS, pt2] = \nu(\lambda_5, BC2_2) = \nu(\lambda_5, AB_2) = \nu(\lambda_5, AC_2) \tag{20}$$

For the genealogy produced by lineage sorting in parent tree 2, $BC1_2$, we have:

$$\nu(\lambda_2, BC1_2) = \nu(\lambda_3, BC1_2) = \frac{1}{1-e^{-(t_2-t_m)}} \int_0^{t_2-t_m} (1 - e^{-\mu(t_m+x)}) e^{-x} dx \tag{21}$$

$$\nu(\lambda_1, BC1_2) = \frac{1}{1-e^{-(t_3-t_2)}} \int_0^{t_3-t_2} (1 - e^{-\mu(t_m+(t_2-t_m)+x)}) e^{-x} dx \tag{22}$$

$$\nu(\lambda_4, BC1_2) = \int_0^{t_2-t_m} \frac{e^{-y}}{1-e^{-(t_2-t_m)}} \left( \int_0^{t_3-t_2} (1 - e^{-\mu((t_2-t_m)-y+x)}) \frac{e^{-x}}{1-e^{-(t_3-t_2)}} dx \right) dy \tag{23}$$

$$\nu(\lambda_5, BC1_2) = \nu(\lambda_5, AB1_1) \tag{24}$$

Finally, we consider parent tree 3. The mutation probabilities have the same formulation as parent tree 2, with two key changes: since parent tree 3 is shorter (*Figure 2* of main text), $t_2$ is replaced by $t_1$. This also applies to the value of $\Lambda$, which we will denote for parent tree 3 as $\Lambda_3 = 1 + \frac{1}{2}e^{-3(t_3-t_1)} - \frac{3}{2}e^{-(t_3-t_1)}$. For the ILS genealogies $BC2_3$, $AB_3$, and $AC_3$, this gives:

$$\nu_1[ILS, pt3] = \frac{1}{\Lambda_3} \int_0^{t_3-t_1} (1 - e^{-\mu(t_m+(t_1-t_m)+x)}) \frac{3}{2}(e^{-x} - e^{-3x}) dx \tag{25}$$

$$\nu_2[ILS, pt3] = \frac{1}{\Lambda_3} \int_0^{t_3-t_1} (1 - e^{-\mu(t_m+(t_1-t_m)+x)}) 3e^{-3x}(1 - e^{-((t_3-t_1)-x)}) dx \tag{26}$$

$$\nu_4[ILS, pt3] = \frac{1}{\Lambda_3} \int_0^{t_3-t_1} 3e^{-3y} \left( \int_0^{(t_3-t_1)-y} e^{-x}(1-e^{-\mu x})dx \right)dy \tag{27}$$

$$\nu_5[ILS, pt3] = \frac{1}{\Lambda_3} \int_0^{t_3-t_1} (1-e^{-\mu((t_3-t_1)-x)})3e^{-3x}dx \tag{28}$$

Where:

$$\nu_1[ILS, pt3] = \nu(\lambda_1, BC2_3) = \nu(\lambda_3, AB_3) = \nu(\lambda_2, BC_3) \tag{29}$$

$$\begin{aligned} \nu_2[ILS, pt3] &= \nu(\lambda_2, BC2_3) = \nu(\lambda_3, BC2_3) = \\ \nu(\lambda_1, AB_3) &= \nu(\lambda_2, AB_3) = \nu(\lambda_1, AC_3) = \nu(\lambda_3, AC_3) \end{aligned} \tag{30}$$

$$\nu_4[ILS, pt3] = \nu(\lambda_4, BC2_3) = \nu(\lambda_4, AB_3) = \nu(\lambda_4, AC_3) \tag{31}$$

$$\nu_5[ILS, pt3] = \nu(\lambda_5, BC2_3) = \nu(\lambda_5, AB_3) = \nu(\lambda_5, AC_3) \tag{32}$$

Finally, for the genealogy $BC1_3$, the mutation probabilities are as follows:

$$\nu(\lambda_2, BC1_3) = \nu(\lambda_3, BC1_3) = \frac{1}{1-e^{-(t_1-t_m)}} \int_0^{t_1-t_m} (1-e^{-\mu(t_m+x)})e^{-x}dx \tag{33}$$

$$\nu(\lambda_1, BC1_3) = \frac{1}{1-e^{-(t_3-t_1)}} \int_0^{t_3-t_1} (1-e^{-\mu(t_m+(t_1-t_m)+x)})e^{-x}dx \tag{34}$$

$$\nu(\lambda_4, BC1_3) = \int_0^{t_1-t_m} \frac{e^{-y}}{1-e^{-(t_1-t_m)}} \left( \int_0^{t_3-t_1} (1-e^{-\mu((t_1-t_m)-y+x)}) \frac{e^{-x}}{1-e^{-(t_3-t_1)}}dx \right)dy \tag{35}$$

$$\nu(\lambda_5, BC1_3) = \frac{1}{1-e^{-(t_3-t_1)}} \int_0^{t_3-t_1} (1-e^{-\mu((t_3-t_1)-x)})e^{-x}dx \tag{36}$$

## 2 When does introgression makes hemiplasy more likely than ILS alone?

The probability of hemiplasy with $C \rightarrow B$ introgression is

$$P_e = (1-\delta)P_e[BC_1] + \delta(P_e[BC1_2] + P_e[BC2_2]) \tag{37}$$

From this, it can be seen that introgression makes hemiplasy more likely than ILS alone when:

$$P_e[BC1_2] + P_e[BC2_2] > P_e[BC_1] \tag{38}$$

When is this true? Substituting the relevant expressions from the main text gives:

$$\begin{aligned} &(1-e^{-(t_2-t_m)})\nu(\lambda_4, BC1_2) \prod_{i \neq 4}(1-\nu(\lambda_i, BC1_2)) + \\ &(\frac{1}{3}e^{-(t_2-t_m)})\nu(\lambda_4, BC2_2) \prod_{i \neq}(1-\nu(\lambda_i, BC2_2)) \\ &> (\frac{1}{3}e^{-(t_2-t_1)})\nu(\lambda_4, BC_1) \prod_{i \neq}(1-\nu(\lambda_i, BC_1)) \end{aligned} \tag{39}$$

$$(1 - e^{-(t_2 - t_m)})v(\lambda_4, BC1_2) \prod_{i \neq 4}(1 - v(\lambda_i, BC1_2)) >$$

$$(\frac{1}{3}e^{-(t_2 - t_1)})v(\lambda_4, BC_1) \prod_{i \neq}(1 - v(\lambda_i, BC_1)) - \qquad (40)$$

$$(\frac{1}{3}e^{-(t_2 - t_m)})v(\lambda_4, BC2_2) \prod_{i \neq}(1 - v(\lambda_i, BC2_2))$$

The mutation probabilities on the right side of the inequality are equal since they are on the same topology with the same branch lengths. Therefore, *Equation 40* can simplified:

$$(1 - e^{-(t_2 - t_m)})v(\lambda_4, BC1_2) \prod_{i \neq 4}(1 - v(\lambda_i, BC1_2)) >$$

$$v(\lambda_4, BC_1) \prod_{i \neq 4}(1 - v(\lambda_i, BC_1))(\frac{1}{3}e^{-(t_2 - t_1)} - \frac{1}{3}e^{-(t_2 - t_m)}) \qquad (41)$$

As the most conservative case, let us assume a hybrid speciation scenario where $t_1 = t_m$. This represents the most conservative introgression scenario, since *Figure 4* in the main text shows that more recent introgression makes hemiplasy more likely. In this case, the right side of the inequality simplifies to 0, leaving

$$(1 - e^{-(t_2 - t_m)})v(\lambda_4, BC1_2) \prod_{i \neq 4}(1 - v(\lambda_i, BC1_2)) > 0 \qquad (42)$$

This is true whenever $t_2 > t_m$, which is true by definition in this model. Therefore, introgression always makes hemiplasy more likely than ILS alone.

