## [Decision Letter]

**Acceptance summary:**

Hibbins and co-authors examine how evolution of binary traits along gene trees that are discordant with the species tree due to incomplete lineage sorting and introgression can generate misleading results for our understanding of the occurrence of homoplasy in cases of convergent evolution. The effects of hemiplasy in trait evolution are well known in the field but have been rarely tested, especially in the context of introgression. The authors show that introgression makes hemiplasy more likely through simulation analyses, develop a software to account for the probability of hemiplasy and homoplasy, and demonstrate its utility using two empirical examples.

**Decision letter after peer review:**

[Editors’ note: the authors submitted for reconsideration following the decision after peer review. What follows is the decision letter after the first round of review.]

Thank you for submitting your work entitled "Determining the probability of hemiplasy in the presence of incomplete lineage sorting and introgression" for consideration by *eLife*. Your article has been reviewed by three peer reviewers, one of whom is a member of our Board of Reviewing Editors, and the evaluation has been overseen by a Senior Editor. The following individual involved in review of your submission has agreed to reveal their identity: Claudia Solis-Lemus (Reviewer #3).

Based on discussions between the reviewers, the Reviewing Editor, and the Senior Editor, and the individual reviews below, we regret to inform you that your work will not be considered further for publication in *eLife*. As you will see from the reviewers' comments below, there was considerable disagreement between reviewers about the novelty and rigor of this work. One of the reviewers felt that the novelty of this work did not meet the bar for *eLife* and two of the reviewers felt that the rigor of this work was not yet sufficient for publication. It was also noted that the revisions requested are not trivial and their outcome not predictable. However, some of the reviewers also felt that *eLife* would be an appropriate venue for this work (if you were able to address the substantial criticisms raised). Given all this, and assuming you are able to address the major issues, *eLife* would be open to considering a re-submission of this work. Please note that this re-submitted article would be treated as a new submission.

The reviewers appreciated that incorporation of ILS+gene flow to the study of hemiplasy would be an important step forward (although, as stated above, they debated how considerable this advance would be). The main concerns of the study were:

1) The example data set used (of the green-blooded lizards) lacks evidence of introgression. It will be essential for the authors to use an example that has evidence of both ILS and introgression. Furthermore, the example should clearly illustrate the advantage of the new ILS+introgression method developed by the authors over the standard ILS-only method.

2) The authors should benchmark their approach. To do this, the authors could simulate under a known history, then infer the network under the best available approaches, then use their method to show the variability in the *HeIST* inference. The authors should similarly allow for uncertainty to propagate in real data.

3) The consequences of the ad-hoc approach to inferring tip lengths should be carefully investigated.

4) Partition the probability of hemiplasy into that attributable to ILS vs Introgression components. Again, bench-marking of these analyses will be necessary.

5) The manuscript's novelty would need to be made more accessible to the broad audience of *eLife* (the reviewers made several suggestions for doing so, e.g., explaining hemiplasy more, incorporating discussion of LGT / HGT, etc.).

*Reviewer #1:*

In this manuscript, Hibbins and co-authors examined how evolution of binary traits along discordant gene trees due to ILS and introgression can generate misleading results for our understanding of the occurrence of homoplasy in cases of convergent evolution. The effects of hemiplasy in trait evolution are well known in the field but have been rarely tested, especially in the context of introgression. The authors have shown that introgression makes hemiplasy more likely through their simulation and also developed the software package to account for the probability of hemiplasy and homoplasy. The authors then apply their tool to explore the evolution of blood color in empirical lizard data and find that the hemiplasy is more likely to explain the previously observed trait incongruence. The manuscript is well written and clear and the algorithm is state-of-art, and it will be of valuable information for the research community studying trait evolution in a variety of organisms.

1) The authors examined the effects of ILS and introgression, but there are other types of the incongruence such as HGT or hidden paralogy that have not mentioned in the manuscript. It will be great to bring these, especially HGT, into the context of this work, acknowledge that they can do cause gene trees to deviate from species trees, and discuss if this method can also handle traits evolved through paralogs or HGT.

2) The model shows that the most important factors contributing to a high risk of hemiplasy relative to homoplasy are short internal branches. Can you quantify this by *HeIST*? For example, you can simulate a same topology with different scales of internal branch lengths to quantify the distribution of the level of hemiplasy. It would be of great of interest to explore at what scale the hemiplasy should be the most likely.

3) While the green-blooded lizard example is certainly interesting, it is not one that has evidence of introgression, so it's at best a "null" example. The authors should use an example of a data set that has evidence of introgression – seems strange that this is their example of choice when the rest of their paper is about integrating both ILS and introgression. Ideally, the authors would find an example whose genetic basis is known so that hemiplasy can be validated by examining the phylogeny of the locus giving rise to the trait. I realize that this may be challenging but the authors' arguments about the likelihood of hemiplasy in the green-blooded lizards are substantially weakened by the fact that the genetic basis of the green-blood trait is not known (and therefore the authors cannot be 100% certain that there was hemiplasy involved).

*Reviewer #2:*

This research has been interested in the potential confusion between "hemiplasy" (a shared trait found in disparate lineages due to ILS and/or introgression), and homosplasmy (a shared trait found in disparate lineages due to independent substitutions). Building on previous work which calculated the relative probability of hemiplasmy under ILS, this work includes introgression as a potential source of hemiplasmy. After presenting some maths for the three taxon case, the authors present a useful helper program "*HeIST*" to use ms to simulate this scenario so that inference can be made for tress with more than three taxa. Because their model is coalescent and requires tip length in coalescent units (which coalescent tree inference methods do not infer), they propose an ad-hoc method in which they regress internal branch length in coalescent units (based on concordance factors) on branch length in substitutions and use this regression to predict coalescent tip branch length from the number of substitutions per site. Finally, the authors use this method to revisit claims of repeated independent evolution of green blood in lizards.

The work is clearly explained. The math in this paper is largely solid, resulting in intuitive and believable results, and would be at home in a theoretical oriented journal like TPB. The simulation framework is the sort of solid ad-hoc simulation expected from empirical papers rather than a "method-development paper". The empirical example is interesting, but does not suit the method. The authors due a great job of listing the many caveats associated with their approach.

Biggest concerns: There is ample opportunity to strengthen this manuscript. Here I highlight the areas which require the most attention. I believe a seriously improved manuscript could be a valuable contribution to a more specialized journal.

1) The relationship between the theory and the data analysis is incredibly weak, and I don't believe this work is publishable with such a disconnect between theory and analysis. The theoretical advance is to incorporate introgression in previous models of the risk of hemiplasy. However, (as far as I can tell) the authors do not model introgression in their inference. As such, it this data set is not well-suited for the methodological development. A data set with introgression would help show why this method is necessary, and will highlight the challenges that arise in applying the method to data that motivated it.

2) Evaluation of model performance. In the Discussion the authors note the many ways that their method can fail. Most of which follow from "garbage in garbage out", if introgression rates and timing, tree inference and timing etc is off, the method will be off too. A stronger method paper would incorporate the imperfection in our ability to know these parameters when presenting a model whose accuracy depends on them. Likewise, the simple regression seems to perform relatively poorly here, and will likely do even worse in the face of gene flow. As such, it would be worthwhile to show how sensitive inference is to mi specified tip lengths. Both of these concerns will have different effects over a range of biological scenarios, so a broad exploration of performance is required.

*Reviewer #3:*

The authors present a novel approach to study convergent traits under both ILS and introgression. They make the distinction between homoplasy and hemiplasy, and study the theoretical probabilities of both scenarios under a multispecies coalescent model on a 3-taxon network. In addition, they produce an open-source software to calculate probabilities of hemiplasy and homoplasy via simulations on larger trees or networks.

The manuscript is extremely well-written, clear and easy to follow. Everything is well-explained. In addition, the material is extremely relevant for the scientific community, and the authors justify every step of their methodology in a transparent manner.

The only major comment is that for someone not familiar already with the concept of hemiplasy, it took me a little bit to understand the distinction to homoplasy. I understand that the authors are building on Guerrero and Hahn, 2018, so having read this paper seems like a prerequisite. However, the authors make a huge effort to make this manuscript self-contained by including a short summary of the findings in Guerrero and Hahn, 2018. I would add perhaps a bit more of the description between hemiplasy and homoplasy found in Guerrero and Hahn (for example, Figure 1) for those readers who might still need a bit more explanation to build intuition

[Editors’ note: further revisions were suggested prior to acceptance, as described below.]

Thank you for submitting your article "Determining the probability of hemiplasy in the presence of incomplete lineage sorting and introgression" for consideration by *eLife*. Your article has been reviewed by three peer reviewers, one of whom is a member of our Board of Reviewing Editors, and the evaluation has been overseen by Patricia Wittkopp as the Senior Editor. The reviewers have opted to remain anonymous.

The reviewers have discussed the reviews with one another and the Reviewing Editor has drafted this decision to help you prepare a revised submission.

Summary:

Hibbins and co-authors examine how evolution of binary traits along gene trees that are discordant with the species tree due to incomplete lineage sorting and introgression can generate misleading results for our understanding of the occurrence of homoplasy in cases of convergent evolution. The effects of hemiplasy in trait evolution are well known in the field but have been rarely tested, especially in the context of introgression. The authors show that introgression makes hemiplasy more likely through simulation analyses, develop a software to account for the probability of hemiplasy and homoplasy, and demonstrate its utility using two empirical examples.

Revisions:

We greatly appreciate the substantive responses to the previous round of comments. We have a couple of additional requests:

1) Biological values for parameters in the empirical example should be provided in the text. i.e. include Ne and mu, rather than there compound parameter, theta, and the parameter choices should be better explained and justified. This is important for evaluation of the results – for example, we are surprised that these lizard and butterflies would have the same value of theta.

2) Evaluation of the performance of *HeIST* based inference should include realistic trade-offs in the estimation of population mutation rates, introgression time, introgression proportion, and split time. Our sense is that any attempt to estimate these parameters from real data would involve some correlations in their inferred values, so it would be good to know how this would impact inference.

---

## [Author Response]

[Editors’ note: the authors resubmitted a revised version of the paper for consideration. What follows is the authors’ response to the first round of review.]

The reviewers appreciated that incorporation of ILS+gene flow to the study of hemiplasy would be an important step forward (although, as stated above, they debated how considerable this advance would be). The main concerns of the study were:1) The example data set used (of the green-blooded lizards) lacks evidence of introgression. It will be essential for the authors to use an example that has evidence of both ILS and introgression. Furthermore, the example should clearly illustrate the advantage of the new ILS+introgression method developed by the authors over the standard ILS-only method.

We have analyzed a new empirical case study involving a chromosomal inversion in *Heliconius*, using data originally published by Edelman et al., 2019. The clade in which this inversion arose has clear evidence of both ILS and introgression, and analyses by Edelman et al. suggested a role for introgression specifically in the origin of the inversion. The results of our analysis confirm their finding that a single origin of the inversion is most likely. See responses below for more specific details.

2) The authors should benchmark their approach. To do this, the authors could simulate under a known history, then infer the network under the best available approaches, then use their method to show the variability in the HeIST inference. The authors should similarly allow for uncertainty to propagate in real data.

We have performed several new analyses and made updates to *HeIST* to address concerns about accuracy and incorporating uncertainty. See responses to reviewers below for detailed comments.

3) The consequences of the ad-hoc approach to inferring tip lengths should be carefully investigated.

We have performed a new analysis of the bias introduced by our approach. Overall, the effect is to artificially increase tree branch lengths, which leads to conservative estimates of the probability of hemiplasy. See below for details.

4) Partition the probability of hemiplasy into that attributable to ILS vs Introgression components. Again, bench-marking of these analyses will be necessary.

*HeIST* now returns a summary of how many focal cases originate from an introgressed history vs. the species history. In addition, users can analyze the same tree in *HeIST* with or without introgression events specified in order to quantify their contribution to overall hemiplasy risk; we do this in our new benchmarking simulations.

5) The manuscript's novelty would need to be made more accessible to the broad audience of eLife (the reviewers made several suggestions for doing so, e.g., explaining hemiplasy more, incorporating discussion of LGT / HGT, etc.).

We have made substantial revisions to the text to address these concerns. See detailed comments below.

Reviewer #1:In this manuscript, Hibbins and co-authors examined how evolution of binary traits along discordant gene trees due to ILS and introgression can generate misleading results for our understanding of the occurrence of homoplasy in cases of convergent evolution. The effects of hemiplasy in trait evolution are well known in the field but have been rarely tested, especially in the context of introgression. The authors have shown that introgression makes hemiplasy more likely through their simulation and also developed the software package to account for the probability of hemiplasy and homoplasy. The authors then apply their tool to explore the evolution of blood color in empirical lizard data and find that the hemiplasy is more likely to explain the previously observed trait incongruence. The manuscript is well written and clear and the algorithm is state-of-art, and it will be of valuable information for the research community studying trait evolution in a variety of organisms.1) The authors examined the effects of ILS and introgression, but there are other types of the incongruence such as HGT or hidden paralogy that have not mentioned in the manuscript. It will be great to bring these, especially HGT, into the context of this work, acknowledge that they can do cause gene trees to deviate from species trees, and discuss if this method can also handle traits evolved through paralogs or HGT.

We have made changes to the Introduction and Discussion to address these concerns. In particular, we make it clear that HGT is included in our definition of introgression. While hidden paralogy does lead to incongruence in the inferred gene tree, it is caused by an error in ortholog identification. Therefore, the true history of traits following such trees will not be discordant with the species tree. We have also clarified this point. The text now reads:

“Gene tree discordance can have multiple sources, including biological causes such as incomplete lineage sorting (ILS), introgression, and horizontal gene transfer, and technical causes such as hidden paralogy or errors in gene tree inference…” (Introduction)

“Horizontal gene transfer, which is more common in prokaryotes, would also require networks that contain reticulation edges spanning very long periods of time” (Discussion)

“Errors common to all phylogenetic methods can be introduced into the user-specified species tree/network at several steps, including errors in ortholog identification…” (Discussion)

2) The model shows that the most important factors contributing to a high risk of hemiplasy relative to homoplasy are short internal branches. Can you quantify this by HeIST? For example, you can simulate a same topology with different scales of internal branch lengths to quantify the distribution of the level of hemiplasy. It would be of great of interest to explore at what scale the hemiplasy should be the most likely.

As part of our new set of benchmarking simulations in *HeIST*, we have included three ILS conditions in which the internal branch length is progressively decreased while other aspects of the tree are held constant. This led to a substantial increase in both the raw probability of hemiplasy, and the probability conditional on observing the specified trait distribution. Figure 5 and describe these new results:

“To evaluate the performance of *HeIST*, we simulated across nine conditions with increasing expected probabilities of hemiplasy, across five different trait mutation rates. The results, shown in Figure 5, confirm the theoretical predictions shown in Figure 4: the probability of hemiplasy increases as a function of decreasing internal branch length (ILS1-ILS3), increasing rate of introgression (INT1-INT3), and more recent introgression (INT4-INT6). The effect of the timing of introgression is weaker than the effect of the introgression rate, also in line with theoretical expectations. These results held true for both the probability conditional on observing the specified trait pattern (Figure 5A) and the raw probability (Figure 5B).”

3) While the green-blooded lizard example is certainly interesting, it is not one that has evidence of introgression, so it's at best a "null" example. The authors should use an example of a data set that has evidence of introgression – seems strange that this is their example of choice when the rest of their paper is about integrating both ILS and introgression. Ideally, the authors would find an example whose genetic basis is known so that hemiplasy can be validated by examining the phylogeny of the locus giving rise to the trait. I realize that this may be challenging but the authors' arguments about the likelihood of hemiplasy in the green-blooded lizards are substantially weakened by the fact that the genetic basis of the green-blood trait is not known (and therefore the authors cannot be 100% certain that there was hemiplasy involved).

As mentioned above, we have included a new analysis of a chromosomal inversion in the *Heliconius erato/sara* clade. While this trait is not a phenotype in the strict sense, the inversion arrangement contains a gene underlying important wing phenotypes. Patterns of gene tree discordance and introgression inferred in Edelman et al., 2019, are clearly suggestive of a hemiplastic origin via introgression, and our analysis in *HeIST* confirms that a single origin of the inversion is most likely. See Figures 1B, 6B, and the new section of the Results entitled “A chromosomal inversion in the *Heliconius erato*/*sara* clade likely has a single origin” for further details.

Reviewer #2:[…]Biggest concerns: There is ample opportunity to strengthen this manuscript. Here I highlight the areas which require the most attention. I believe a seriously improved manuscript could be a valuable contribution to a more specialized journal.1) The relationship between the theory and the data analysis is incredibly weak, and I don't believe this work is publishable with such a disconnect between theory and analysis. The theoretical advance is to incorporate introgression in previous models of the risk of hemiplasy. However, (as far as I can tell) the authors do not model introgression in their inference. As such, it this data set is not well-suited for the methodological development. A data set with introgression would help show why this method is necessary, and will highlight the challenges that arise in applying the method to data that motivated it.

We have included a new analysis of a *Heliconius* dataset with introgression (see responses to reviewer 1 for details), in addition to performing extensive test simulations over multiple introgression conditions (see comments below), to address these concerns. We think that the analysis of this dataset does help to show why our method is necessary, and we thank the reviewer for the suggestion.

2) Evaluation of model performance. In the Discussion the authors note the many ways that their method can fail. Most of which follow from "garbage in garbage out", if introgression rates and timing, tree inference and timing etc is off, the method will be off too. A stronger method paper would incorporate the imperfection in our ability to know these parameters when presenting a model whose accuracy depends on them. Likewise, the simple regression seems to perform relatively poorly here, and will likely do even worse in the face of gene flow. As such, it would be worthwhile to show how sensitive inference is to mi specified tip lengths. Both of these concerns will have different effects over a range of biological scenarios, so a broad exploration of performance is required.

We have performed several new analyses that we believe quantify the effects of various kinds of uncertainty on the accuracy of inferences in *HeIST*:

First, we have performed a new set of extensive benchmarking simulations, across nine different simulated conditions (six involving introgression) and five different mutation rates. While primarily intended to demonstrate how variation in true parameters affects the probability of hemiplasy, they can also be interpreted as showing the expected deviation in the probability of hemiplasy if parameters are mis-specified. See Figure 5—figure supplement 4, the new Materials and methods section entitled “Accuracy of *HeIST*”, and the new Results section entitled “*HeIST* effectively captures the effects of ILS and introgression on hemiplasy risk” for a detailed description of these simulations and results.

Second, we have included an analysis that examines how a typical pipeline for generating an input to *HeIST* from an empirical phylogeny can lead to inaccurate inferences. This involved simulating data from a known tree, building a new tree from the simulated data in *RAxML*, estimating site concordance factors, and using that information as input to our regression module followed by tree smoothing. Overall, the effect is to increase both internal and tip branch lengths relative to the known tree, which makes hemiplasy inferences more conservative. See Figure 5—figure supplement 6, the new Materials and methods section entitled “Inferring the tip branch lengths of a phylogeny in coalescent units”and the same new Results section as above for an overview of the simulations and results.

Finally, we have now updated *HeIST* so that it can also use the upper and lower bounds of the 95% confidence interval of predicted tip branch lengths, in addition to the predictions themselves. This allows users to quantify the uncertainty in their results due to mis-specified branch lengths.

Reviewer #3:The authors present a novel approach to study convergent traits under both ILS and introgression. They make the distinction between homoplasy and hemiplasy, and study the theoretical probabilities of both scenarios under a multispecies coalescent model on a 3-taxon network. In addition, they produce an open-source software to calculate probabilities of hemiplasy and homoplasy via simulations on larger trees or networks.The manuscript is extremely well-written, clear and easy to follow. Everything is well-explained. In addition, the material is extremely relevant for the scientific community, and the authors justify every step of their methodology in a transparent manner.The only major comment is that for someone not familiar already with the concept of hemiplasy, it took me a little bit to understand the distinction to homoplasy. I understand that the authors are building on Guerrero and Hahn, 2018, so having read this paper seems like a prerequisite. However, the authors make a huge effort to make this manuscript self-contained by including a short summary of the findings in Guerrero and Hahn, 2018. I would add perhaps a bit more of the description between hemiplasy and homoplasy found in Guerrero and Hahn (for example, Figure 1) for those readers who might still need a bit more explanation to build intuition

We have added a new panel to Figure 3 to provide a visual contrast of homoplasy and hemiplasy, as well as updating the caption of the figure to help describe the distinction in more detail:

“Homoplasy can happen on any gene tree, as long as there are two independent mutations on tip branches (panel A). Homoplasy can also happen via a mutation in the ancestor of all three species, followed by a reversal”

[Editors’ note: what follows is the authors’ response to the second round of review.]

Revisions:We greatly appreciate the substantive responses to the previous round of comments. We have a couple of additional requests:1) Biological values for parameters in the empirical example should be provided in the text. i.e. include Ne and mu, rather than there compound parameter, theta, and the parameter choices should be better explained and justified. This is important for evaluation of the results – for example, we are surprised that these lizard and butterflies would have the same value of theta.

We have added clarifying statements to the Materials and methods section to justify our choices of parameters:

“While specific parameter estimates are not available for this system, our choice of θ reflects broad estimates of N_e_ and µ on the order of 10^5^ – 10^6^ (Lynch, 2006) and 10^-8^ – 10^-9^ per-base per-generation (Lynch 2010), respectively, in vertebrates (see Discussion).”

“That our choice of θ for this system is the same as in our lizard analysis is just a coincidence: it reflects a trade-off between the generally higher effective population size for insects (Lynch, 2006) and the lower mutation rate expected for chromosomal inversions; see Discussion.”

2) Evaluation of the performance of HeIST based inference should include realistic trade-offs in the estimation of population mutation rates, introgression time, introgression proportion, and split time. Our sense is that any attempt to estimate these parameters from real data would involve some correlations in their inferred values, so it would be good to know how this would impact inference.

We acknowledge that parameter inference for phylogenetic networks is often difficult and that many different sets of parameters may be consistent with patterns in observed data. To address this concern, we have performed a new set of simulations across five conditions in which the probability and recency of introgression were increased while holding constant the probability of the gene tree leading to hemiplasy. We found an effect of the timing of introgression on the probability of hemiplasy that is independent of the rate of discordance, emphasizing that parameter inference should be done using gene trees with branch lengths.

This analysis is described in new paragraphs in the Materials and methods and Results. New Figure 5—figure supplements 4 and 5 visualize the choice of parameters and simulation results, respectively. The new paragraph in the Results reads:

“When the parameters of a phylogenetic network are estimated from empirical data, it is possible that many different parameter combinations may be equally likely, especially when only a subset of features are used to fit the model. However, these combinations may differentially affect the probability of hemiplasy: for instance, if the frequency of gene trees is used to fit the network model, but the length of gene tree branches is ignored. To investigate this, we applied *HeIST* to five simulated conditions in which the probability and recency of introgression were increased while the frequency of the discordant gene tree that could cause hemiplasy was held constant (Figure 5—figure supplement 4). We found that, despite a constant gene tree probability, the conditional probability of hemiplasy increased in each successive condition as introgression became more recent and frequent (Figure 5—figure supplement 5). These results to some extent merely serve to reinforce the notion that introgression has an effect on hemiplasy above and beyond the effect of ILS alone: by lengthening the branch on the discordant tree that hemiplastic mutations can occur on, introgression has a larger effect than ILS alone. But even when network models that include introgression are used, the estimated effects on hemiplasy will be conservative if parameters are estimating using gene tree frequencies alone.”